# Multi-View Representation Learning
# via Total Correlation Objective

**HyeongJoo Hwang, Geon-Hyeong Kim, Seunghoon Hong, Kee-Eung Kim**
KAIST
{hjhwang, ghkim}@ai.kaist.ac.kr, {seunghoon.hong, kekim}@kaist.ac.kr

## Abstract

Multi-View Representation Learning (MVRL) aims to discover a shared representation of observations from different views with the complex underlying correlation. In this paper, we propose a variational approach which casts MVRL as maximizing the amount of total correlation reduced by the representation, aiming to learn a shared latent representation that is informative yet succinct to capture the correlation among multiple views. To this end, we introduce a tractable surrogate objective function under the proposed framework, which allows our method to fuse and calibrate the observations in the representation space. From the information theoretic perspective, we show that our framework subsumes existing multi-view generative models. Lastly, we show that our approach straightforwardly extends to the Partial MVRL (PMVRL) setting, where the observations are missing without any regular pattern. We demonstrate the effectiveness of our approach in the multi-view translation and classification tasks, outperforming strong baseline methods.

## 1 Introduction

Multi-View Representation Learning (MVRL) aims to learn a shared representation of multiple observations from different types of views. In MVRL, it is important to encourage the shared representation to be complete enough to capture the correlation across views without losing view-specific information so that the learned representation can be readily applied to rich set of downstream tasks. For example, in sensor fusion for the multi-sensor system [51] and in clinical diagnosis based on patients' various types of medical records [47, 49], one would like to aggregate all the information from various observations in order to uncover the true underlying factors under the correlation. Although using views as many as possible seems to be always beneficial for the task of learning a good representation, it can make the problem itself harder depending on the complexity of correlations across all views and the scalability of handling a number of views.

MVRL becomes even more challenging when the model does not always have access to complete observations from all views for every data instance during training, which we call Partial Multi-View Representation Learning (PMVRL). PMVRL is closer to the practical setting since it is unrealistic to expect that observations from all different kinds of views are always available. For example, it is unlikely for all the sensors in a sensor-based system to have the same frequency to update their measurements or for all kinds of medical records to be available for any patient.

Considering those difficulties, any desirable MVRL methods are encouraged to satisfy three desiderata. The first one is *scalability* to the number of input views. The method should handle multiple observations in a way computationally scalable to arbitrary many views in both training and testing time. In addition, the method needs to be robust to *partial observability*. The method should be able to combine any combination of available observations in the representation space in test time. This is important for PMVRL because we have to handle observations arbitrarily missing not only in testing data but also in training data. Lastly, the method should discover *cross-view association*,

35th Conference on Neural Information Processing Systems (NeurIPS 2021).

which can be learned by identifying both shared and view-specific factors of variation of each view in the representation space. Associating views correctly in the representation space allows the method to utilize any additional observations from different views to improve identification of the shared factors without determining any unobserved ones.

Combining multiple of Variational Auto-Encoder (VAE) [22] for each view[1], several generative models [30, 33, 34, 46] have been recently proposed to address MVRL. Since naive optimization of the evidence lower bound (ELBO) on the joint likelihood of multiple views does not address any of the desiderata, those methods impose different structural bias on the joint representation encoders whose strengths and weaknesses are complementary (see Section 2.4). Although they showed encouraging performance on multiple tasks such as predicting common attributes and inferring missing views, they often fail to simultaneously capture both the shared and view-specific factors of variation and turn out to poorly associate views in our experiments.

In this paper, we address the problem of MVRL with a principled approach grounded in information theory. Specifically, we formulate the representation learning task as maximizing the reduction in Total Correlation [12, 38, 39]. Based on our formulation, we derive a novel objective function that not only offers tractable optimization but also introduces multiple types of variational information bottlenecks which successfully associate views. We then show that our method naturally extends to the PMVRL setting via inverse variance weighting, a classical approach used in sensor fusion. We demonstrate the validity and effectiveness of our method in the multi-view translation and classification downstream tasks. Our contributions are three-fold:

1. Measuring the informativeness of the multi-view representation by Total Correlation, we propose a general information-theoretic framework to learn a complete representation, which encompasses existing multi-view generative models.

2. Under the proposed framework, we identify drawbacks of optimizing ELBO and derive a novel objective function that resolves them. Specifically, our method yields a representation that correctly associates views by capturing not only the common factors of variation but also the view-specific ones.

3. We conducted extensive evaluation with comparing methods and ours in translation and classification tasks in both MVRL and PMVRL settings, showing that our method is the most reliable method to obtain the latent representation agnostic to the downstream tasks.

## 2   Approach

Let $\vec{o} = \{o_v\}_{v=1}^{V}$ be the observation of a data instance composed of $V$ different views, which is sampled from an unknown joint distribution $p_D(\vec{o})$, where we emphasize that this is a data distribution using the subscript $D$. Given these observations, MVRL is the task of learning a *complete* representation $z$ across the views $\vec{o}$. Following [50], we define a complete representation as follows.

**Definition 1 (Completeness for Multi-View Representation [50])** *A multi-view representation $z$ is complete if each observation, i.e., $o_v$ from $\vec{o}$, can be reconstructed from a mapping $f_v(\cdot)$, i.e., $o_v = f_v(z)$.*

The definition directly indicates that a complete representation describes all factors of variations in $\vec{o}$, since every view can be reconstructed solely from the complete representation. While MVRL considers complete observations for training data, PMVRL assumes otherwise, i.e. some views being missing in the training data. This poses a unique challenge of (1) learning to produce the complete representation with partial views (2) whose availability varies per instance in training and testing.

We first relate informativeness of the representation measured in terms of Total Correlation (TC) and the goal of MVRL (Section 2.1). We then show that the TC-based MVRL objective function encompasses existing multimodal generative models and analyze its limits based on a straightforward variational lower bound (Section 2.2). To resolve those limits, we derive an alternative variational lower bound which suits better for MVRL (Section 2.3). Lastly, we finalize our formulation by proposing the representation aggregation model that naturally extends to PMVRL (Section 2.4).

---

[1]We follow the conventional terminology in the related literature [11, 14, 24, 41, 44, 47, 48, 50, 53], but any choice among views, modalities [30, 33, 34, 46], and domains [19] can be suitable for our paper and our baseline methods, as long as there is a common latent representation that explains different views / modalities / domains.

## 2.1 Multi-View Representation Learning with Total Correlation

A complete representation $Z$ should be informative enough to explain the correlation among $V$ different views. Total correlation (TC) [42], defined as the Kullback-Leibler divergence of the joint distribution from the factored marginals, measures the amount of information shared among a finite set of random variables. In our MVRL context, TC is defined as

$$TC(\vec{O}) \triangleq D_{KL}\left[p_D\left(\vec{o}\right) \| \prod_{v=1}^{V} p_D\left(o_v\right)\right]. \tag{1}$$

We aim to find the encoder $p_\theta\left(z|\vec{o}\right)$ such that the knowledge of $z$ would reduce TC as much as possible. This can be formulated by maximizing the objective

$$TC_\theta(\vec{O}; Z) \triangleq TC(\vec{O}) - TC_\theta(\vec{O} \mid Z), \tag{2}$$

where the conditional TC in the last term is given by

$$TC_\theta(\vec{O}|Z) \triangleq \mathbb{E}_{p_\theta(z)}\left[D_{KL}\left[p_\theta\left(\vec{o}|z\right) \| \prod_{v=1}^{V} p_\theta\left(o_v|z\right)\right]\right], \tag{3}$$

which is the expected Kullback-Leibler divergence of the joint conditional from the factored conditionals. The parameterized distributions in the above formula involve the encoder $p_\theta\left(z|\vec{o}\right)$ as follows: $p_\theta\left(z\right) = \int p_\theta\left(z|\vec{o}\right) p_D\left(\vec{o}\right) d\vec{o}$, $p_\theta\left(\vec{o}|z\right) = p_\theta\left(z|\vec{o}\right) p_D\left(\vec{o}\right)/p_\theta\left(z\right)$, and $p_\theta\left(o_v|z\right) = \int p_\theta\left(\vec{o}|z\right) d\vec{o}_{\backslash v}$.

Intuitively, minimization of Eq. (3) (*i.e.*, maximization of Eq. (2)) suits well for MVRL, since (1) any complete representation $Z$ would minimize Eq. (3) (*e.g.*, $Z = \vec{o}$), and (2) it accords with the theoretical result that complete representation $z$ should factorize the generative distribution [44, 50].

Further decomposition of (2) reveals that it encourages $Z$ to encode the correlation across views [12], a desirable property for MVRL:

$$TC_\theta(\vec{O}; Z) = \sum_{v=1}^{V} I_\theta\left(O_v; Z\right) - I_\theta(\vec{O}; Z). \tag{4}$$

The first term in Eq. (4), Mutual Information (MI) between each observation and the representation, enforces the representation to be informative for every observation. On the other hand, the second term takes the role of Information Bottleneck (IB), which encourages the encoder to learn minimal sufficient representation [37]. Consequently, simultaneous optimization of both terms naturally allows the representation $Z$ to capture correlations among views while being minimally sufficient. Note that when $V$=2, Eq. (4) coincides with Interaction Information [4, 19, 25, 36], which quantifies the amount of information shared among two views and their joint representation (see Section A.4).

## 2.2 Limitations of VAE models for multi-view data

Unfortunately, a direct optimization of MIs in Eq. (4) is intractable [2, 12]. A straightforward approach would be employing approximate distributions $q_\phi^v\left(o_v|z\right) \approx p_\theta\left(o_v|z\right)$ and $r\left(z\right) \approx p_\theta\left(z\right)$, and optimize a variational lower bound of Eq. (4) as follows (see Section A.1 in the supplementary material for full derivation):

$$TC_\theta(\vec{O}; Z) \geq \sum_{v=1}^{V}\left[H\left(O_v\right) + \mathbb{E}_{p_\theta(z|\vec{o})p_D(\vec{o})}\left[\ln q_\phi^v\left(o_v|z\right)\right]\right] - \underbrace{\mathbb{E}_{p_D(\vec{o})}\left[D_{KL}\left[p_\theta(z|\vec{o})\|r(z)\right]\right]}_{\text{VIB}}, \tag{5}$$

where the entropy terms can be dropped from optimization since they are determined by the true data distribution $p_D\left(\vec{o}\right)$. Without the entropy terms, we note that Eq. (5) is essentially identical to evidence lower bound (ELBO) of the variational auto-encoder (VAE) models for multi-view data [30, 33, 34, 46] by switching the notations, for $p$ for encoder and $q$ for decoder. This notation switch follows the convention in [2, 12].

Although this lower bound contains the variational information bottleneck term that encourages the representation to be minimally sufficient for generalizing well even with small training data [2], it has the following fundamental limitations:

1. **Unbalanced representation**: If a subset of views is overwhelmingly informative enough to reconstruct the others, the encoder $p_\theta(z|\vec{o})$ may learn to rely on those views while ignoring the rest, yielding a degenerate solution of Eq. (4) that fails cross-view association. This is problematic in MVRL since such views may not be available at test time.

2. **Missing views**: The encoder $p_\theta(z|\vec{o})$ requires complete observations $\vec{o}$ not only in training but also in testing phases, while MVRL requires the model to encode incomplete observations $\tilde{o} \subseteq \vec{o}$ in test time. Furthermore, PMVRL requires to handle incomplete observations even in training.

In order to overcome these challenges, prior methods impose special structures (i.e. inductive hypotheses) for $p_\theta(z|\vec{o})$, such as Product of Experts (PoE) [17, 46], Mixture of Experts (MoE) [30], or Mixture of Product of Experts (MoPoE) [34]. In our work, we present a more principled approach to this problem by deriving an alternative lower bound, described in the next section.

## 2.3 Conditional Variational Information Bottleneck

To resolve the first issue raised in the previous section, we start from Eq. (4) and reformulating it as follows:

$$
\begin{aligned}
TC_\theta(\vec{O}; Z) &= \sum_{v=1}^{V} \left[ \frac{V-1}{V} I_\theta(O_v; Z) + \frac{1}{V} I_\theta(O_v; Z) - \frac{1}{V} I_\theta\left(\vec{O}; Z\right) \right] \\
&= \frac{1}{V} \sum_{v=1}^{V} \left[ (V-1) I_\theta(O_v; Z) - I_\theta(\vec{O}_{\setminus v}; Z|O_v) \right],
\end{aligned}
\tag{6}
$$

where the last equality is due to the chain rule of MI (see Section A.5 in the supplementary material for details). Interestingly, Eq. (6) transforms IB in Eq. (4) into multiple *conditional* MIs between the latent representation and $V-1$ other views given every view, each of which penalizes the extra information of the representation *not inferable* from the given view. Although Eq. (6) is essentially equal to Eq. (4), its conditional information constraints give us intuition to derive a new tractable lower bound on $TC_\theta(\vec{O}; Z)$ that regularizes unbalanced representation, which we present below.

Since the conditional MIs in Eq. (6) involve $p_\theta(z|o_v) = \int p_\theta(z|\vec{o}) p_D(\vec{o}_{\setminus v}|o_v) d\vec{o}_{\setminus v}$ which requires to compute intractable integration, we use the variational upper bound of those terms by introducing approximate distributions $r_\psi^v(z|o_v) \approx p_\theta(z|o_v)$ as follows (see Section A.2 in the supplementary material for the full derivation and analysis):

$$
\begin{aligned}
TC_\theta(\vec{O}; Z) \geq &\frac{V-1}{V} \sum_{v=1}^{V} \left[ H(O_v) + \mathbb{E}_{p_\theta(z|\vec{o}) p_D(\vec{o})} \left[ \ln q_\phi^v(o_v|z) \right] \right] \\
&- \frac{1}{V} \sum_{v=1}^{V} \underbrace{\mathbb{E}_{p_D(\vec{o})} \left[ D_{KL} \left[ p_\theta(z|\vec{o}) \| r_\psi^v(z|o_v) \right] \right]}_{\text{Conditional VIB}}.
\end{aligned}
\tag{7}
$$

This lower bound is equipped with conditional VIBs which provide a number of benefits over Eq. (5) in handling challenges mentioned in the previous section. First, conditional VIBs, which upper bounds condtional MIs in Eq. (6) by introducing the view-specific encoder $r_\psi^v(z|o_v)$ for each view, regularize $p_\theta(z|\vec{o})$ to encode representation *inferable* from $r_\psi^v(z|o_v)$ of every view. Consequently, the joint representation is enforced to be balanced rather than to be prone to uneven dependency on some subset of views. Second, each of them uses *forward* KL divergence $D_{KL}[p_\theta(z|\vec{o})\|r_\psi^v(z|o_v)]$ to calibrate each encoder $r_\psi^v(z|o_v)$ to the joint encoder $p_\theta(z|\vec{o})$, encouraging $r_\psi^v(z|o_v)$ to cover all the supports or modes of $p_\theta(z|\vec{o})$. As a consequence, one can extract the representation $z$ even when some views are missing in the observation. This property is critically important in (P)MVRL, where one needs to infer the complete representation from the partially available views without being overly confident on any unobserved factors. We remark that mmJSD [33] adopts *reverse* KL divergence, which is not ideal as we later demonstrate in the experiments.

Finally, although Eq. (7) has aforementioned desirable properties for MVRL, it is prone to overfitting when the size of training data is limited. This is because $r_\psi^v$ can optimize the conditional VIB by

simply memorizing instead of learning to infer the representation of $p_\theta(z|\vec{o})$. In order to prevent overfitting, we found that VIB in Eq. (5) is an effective regularization as it favors the minimal sufficient encoding of the representation, which will be demonstrated in Section 4.2.1. Therefore, we formulate our objective function as a convex combination of Eq. (5) and Eq. (7) so that we regularize the training via VIB (see Section A.3 in the supplementary file for full derivation):

$$TC_\theta(\vec{O};Z) \geq \frac{V-\alpha}{V} \sum_{v=1}^{V} \left[ H\left(O_v\right) + \mathbb{E}_{p_\theta(z|\vec{o})p_D(\vec{o})} \left[\ln q_\phi^v\left(o_v|z\right)\right]\right]$$

$$- \frac{\alpha}{V} \sum_{v=1}^{V} \underbrace{\mathbb{E}_{p_D(\vec{o})} \left[D_{KL}\left[p_\theta(z|\vec{o}) \| r_\psi^v(z|o_v)\right]\right]}_{\text{Conditional VIB}} - (1-\alpha) \underbrace{\mathbb{E}_{p_D(\vec{o})}\left[D_{KL}\left[p_\theta(z|\vec{o})\|r(z)\right]\right]}_{\text{VIB}}, \quad (8)$$

where $\alpha$ is the hyperparmeter that trades off learning minimal sufficient representation in favor of calibrating $r_\psi^v$. For simplicity, we model the encoder, decoder, and approximate marginal distributions using the parameterized Gaussians with the diagonal covariance matrix, i.e. $r_\psi^v\left(z|o_v\right) = N\left(\mu_v, \sigma_v^2 I\right)$, $q_\phi^v\left(o_v|z\right) = N\left(\hat{\mu}_v, I\right)$, and $r\left(z\right) = N(0, \mathrm{I})$, respectively.

While the view-specific encoders $r_\psi^v$ allow us to extract the representation from any available view individually, combining any subset of these representations (fusion) still remains as a problem. In addition, the use of joint-view encoder $p(z|\vec{o})$ makes our method limited to complete observations $\vec{o}$ during training, which needs to be addressed for PMVRL. In the next section, we show that these issues can be effectively resolved by a simple model design for $p_\theta\left(z|\vec{o}\right)$.

## 2.4 Models for Joint Representation Encoder

We review the models adopted by prior methods that make the joint representation encoder $p_\theta\left(z|\vec{o}\right)$ amenable to missing views and discuss their strengths and weaknesses.

**PoE** Product-of-Experts (PoE) [17] combines multiple probability distributions by their product. MVAE [46] models the joint representation encoder as a PoE, treating view-specific encoders as experts. The PoE can produce a sharper distribution as we increase the number of input views, thus an effective method for aggregating information across any subset of view-specific encoders. Assuming each of view-specific encoders as Gaussian distributions such that $r_\psi^v\left(z|o_v\right) = N\left(\mu_v, \sigma_v^2 I\right)$, the PoE joint encoder is obtained with computation linearly scales to the number of views by

$$p_\theta\left(z|\vec{o}\right) \triangleq N\left(\mu_p, \sigma_p^2 \mathbf{I}\right), \quad \text{where} \quad \mu_p \triangleq \frac{\sum_{v=1}^{V} \mu_v/\sigma_v^2}{\sum_{v=1}^{V} 1/\sigma_v^2} \quad \text{and} \quad \sigma_p^2 \triangleq \frac{1}{\sum_{v=1}^{V} 1/\sigma_v^2}. \quad (9)$$

Eq. (9) is also the formula of Inverse-Variance Weighted (IVW) method [6, 7], a classical method in statistics for aggregating multiple random variables, such as sensor fusion.

Unfortunately, a naive application of the PoE to the ELBO formulation (Eq.(5)) may fail to optimize the individual encoders, which is important in learning the balanced representation. MVAE [46], as an example, randomly samples subsets of views among $2^V$ combinations and jointly optimizes their ELBOs in order to ensure that all the view-specific encoders are optimized under PoE. However, such treatment may result in a precision miscalibration of view-specific encoders [30].

**MoE** The Mixture-of-Expert (MoE) takes an arithmetic mean of probability distributions, which is computationally scalable to the number of views as well. MMVAE [30] and mmJSD [33] adopt MoE of $r_\psi^v(z|o_v)$ as the model for the joint representation encoder. In MMVAE, the MoE is trained by pair-wise optimization in such a way that the latent representation from a view-specific encoder can reconstruct the observation in other views as well as its own view. However, this does not necessarily imply that the latent representation successfully aggregates the information across views. mmJSD addresses this issue by adopting a common learnable prior across views.

We remark that, in Eq. (8), modeling $p_\theta\left(z|\vec{o}\right)$ as MoE of $r_\psi^v(z|o_v)$ and setting $\alpha = 0$ yields the ELBO objective version of MMVAE. Assuming the same model for $p_\theta\left(z|\vec{o}\right)$ and $\alpha = 0$, and modeling $r(z)$ as the PoE of $r_\psi^v(z|o_v)$ yields the objective function of mmJSD. For tractable optimization of the KL term involving MoE, mmJSD derives a lower bound on the ELBO by decomposing it into multiple KL terms. This bound can be also obtained from Eq. (8) by setting $\alpha = 1$ and using the *reverse* KL for the conditional VIB terms. However, we empirically show that minimization of the reverse KL is not helpful with learning complete representation in Section 4.

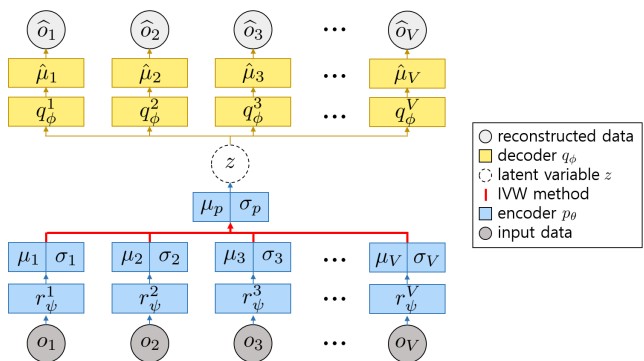

Figure 1: The architecture of Multi-View Total Correlation Auto-Encoder.

**MoPoE** The Mixture-of-Product-of-Experts (MoPoE) is a mixture of $2^V$ combination of PoE experts, which is used as the model for joint encoder in MoPoE-VAE [34]. Since the MoPoE joint encoder takes into account all possible combinations of views, it naturally learns to aggregate information across any given views while optimizing every view-specific encoder. Similar to mmJSD, MoPoE-VAE derives a lower bound on the objective to decompose the KL term into $2^V$ KL terms with analytic solutions. However, this would render the method intractable for tasks with many views.

**Multi-View Total Correlation Auto-Encoder** Since our model uses conditional VIBs that explicitly calibrate all the representations encoded by view-specific encoders, we can safely choose PoE as the model for the joint representation encoder without suffering from the precision miscalibration. We call our resulting model the Multi-View Total Correlation Auto-Encoder (MVTCAE), depicted in Figure 1. Thanks to PoE joint representation encoder, MVTCAE linearly scales to the number of input views. Furthermore, when training with partial observations, MVTCAE simply treats the covariance matrix of $q_\phi^v(o_v|z)$ to be $\infty \mathbf{I}$ for any missing observation $o_v \notin \tilde{o}$, so that it does not contribute to the reconstruction loss for missing views, similar to [50]. As a result, MVTCAE can be naturally extended to PMVRL. We show that MVTCAE successfully associates views in Section 4.

## 3 Related Work

**Information-Theoretic Representation Learning** Information Bottleneck (IB) [37] was introduced as a regularization method to obtain minimal sufficient encoding by constraining the amount of information captured by the latent variable from the observed variable. Deep Variational IB (VIB) [2] extended IB by parameterizing it with a neural network, which results in a simple yet effective method to achieve a representation that generalizes well. Furthermore, using VIB in unsupervised learning has been revealed to have close relationships among VIB, VAE [22] and $\beta$-VAE [16]. A number of follow-up works [27, 31] propose encouraging the encoders to learn representation invariant to any attribute given in advance. Similarly, a modified version of VIB was introduced [11] for learning a view-invariant representation across two views. While IB and VIB are concerned with computing MI, two similar but distinct generalizations of MI have been applied to learning disentangled representation, which are TC [42] and Interaction Information (II) [4, 25, 36]. The TC quantifies the dependency among all dimensions of the single latent variable, which motivated many works that learn disentangled representation in a single view [5, 10, 12, 20, 21]. II was used for disentangling shared representation from view-specific representation in cross-views [19].

**Multi-View Representation Learning** Canonical Correlation Analysis (CCA) [18] and its variants [1, 3, 41] are classical approaches for unsupervised cross-view representation learning. CCA projects two different views into one common latent space in a way that those two views are maximally correlated in the latent space. KCCA [1] uses kernels and DCCA [3] uses neural networks to learn the common representation. Similarly, DCCAE [41] trains autoencoders to obtain common representations. More recently, a number of notable MVRL methods have been proposed to support more than 2 views. DMF-MVC [53] extracts a common representation of multiple views through deep matrix factorization. MDcR [48] maps each view to a lower-dimensional space and applies the kernel matching to regularize the dependence across multiple views. ITML [9] learns a Mahalanobis distance function by Bregman optimization, whereas LMNN [43] learns a Mahalanobis distance metric to optimize the k-nearest neighbors classifier using labeled data. CPM-Nets [50] gives a formal definition of complete representation in PMVRL and proposes to learn it without encoders.

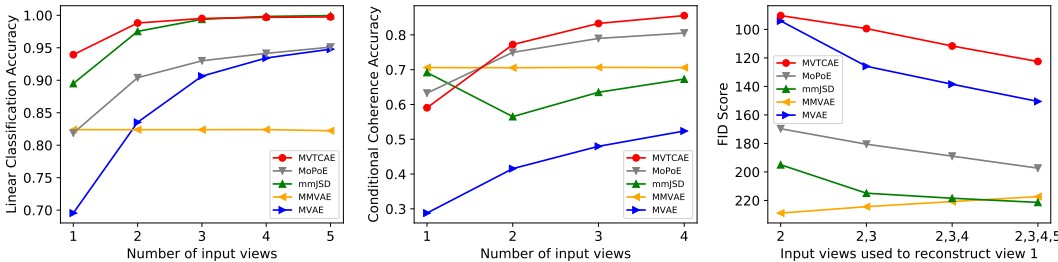

Figure 2: Performance evaluation on the Poly MNIST dataset.

**Multi-View Generative Models (VAEs)**    MVAE [46] and its variants [30, 33, 34] are multi-view generative models that learn shared representation by maximizing the log-likelihood of joint views via latent variables. We compared these methods to ours in Section 2.2 and 2.4.

# 4    Experiments

## 4.1    Multi-View Representation Learning

To verify that our method successfully learns complete representation capturing both common factors and view-specific factors, we evaluate our method on the following two datasets used as evaluation benchmarks in the MVRL literature.

### 4.1.1    Multi-View Classification / Translation on PolyMNIST

We employ PolyMNIST dataset [34] composed of tuples with 5 different MNIST images, which have different backgrounds and writing style but share the same digit label. The background of each view is randomly cropped from one image which is not used by other views. Thus, the digit identity is the common factor of variation while the background and writing style are view-specific factors. There are 60K tuples of training samples and 10K of test samples. Although the digit ID should be observable in any view, it is hard to identify in some images depending on the background and the writing style. Therefore, aggregating information across views is essential for predicting the label.

**Evaluation protocol**    To evaluate the learned representation, we follow the protocol in [34]. Specifically, after training all the models in an unsupervised manner using complete observations, we evaluate the learned representation with three different metrics, which are linear classification accuracy and conditional coherence accuracy. We also evaluate the quality of conditional generation with FID score [15]. We compare our method with various state-of-the-art multi-view generative models which are MVAE [46], MMVAE [30], mmJSD [33], and MoPoE-VAE [34].

To apply our method to classification, we fix the encoders and train a linear classifier to predict labels using the joint representation extracted from $p_\theta(z|\vec{o})$ feeding complete observations in training set. We then classify the representations of all subsets and compute the average classification accuracy over all subsets with the same subset size.

To measure the conditional coherence accuracy, we extract the representation of every subset of views using $p_\theta$ and generate views that are absent in the subset using $q_\phi$. Those generated views are fed into the pretrained CNN-based classifier and see if the prediction from the classifier matches the label of the given subset. The results are averaged over all subsets with the same size.

Finally, we evaluate the sample quality of the first view images generated from different combination of input views ({2}, {2,3}, {2,3,4}, {2,3,4,5}) in terms of FID score. Since FID compares statistics of two sets (one is the set of samples generated by the models and the other is the first view images in the training set in our context), it takes into account not only the quality of generated samples but also the diversity of them. Thus, unlike two previous evaluation metrics, marking a lower (better) FID score requires the model to learn the representation that captures not only the common factors of variation but also view-specific factors to express diversity within the view.

**Results**    The left plot in Figure 2 shows the result of linear classification. Using the PoE as a joint representation encoder same as MVAE, our method clearly outperforms all the baseline methods in the classification task, reaching 94% and 99% accuracy even when only one view and two views are given respectively. Comparing to MVAE, this result implies that conditional VIBs in our method calibrate every view-specific encoders according to the information shared across views so that each of them successfully captures the digit identity, resolving the issue of unbalanced representation.

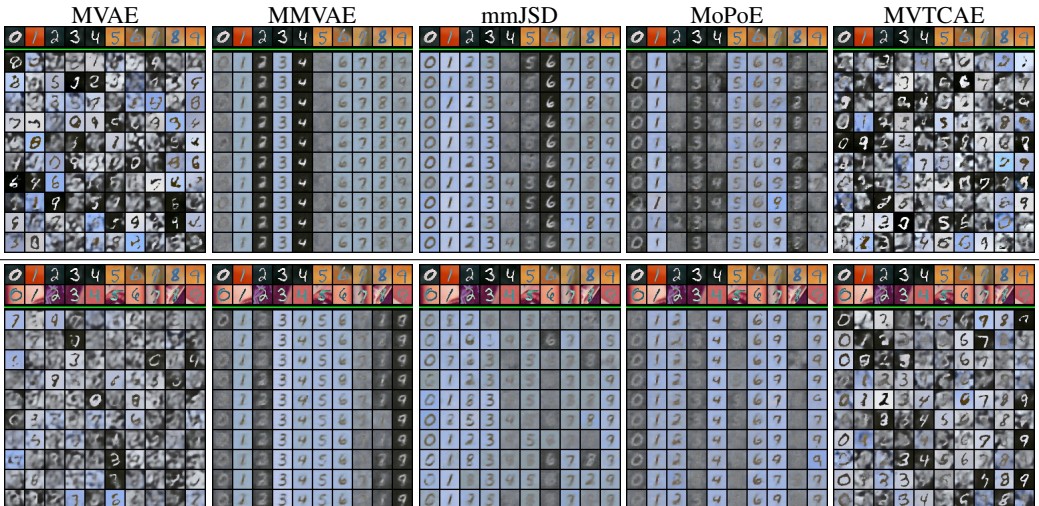

Figure 3: Conditionally generated images of the view 1 given images from the view 2 (top row) and images from the views 2 and 3 (bottom row).

On the other hand, while all the methods show monotonic improvement on its performance as the number of given views increases, the accuracy of MMVAE does not show any noticeable change. It is remarkable that our method even outperforms mmJSD and MoPoE-VAE with simpler aggregation model for the joint encoder, although the two prior methods use both PoE and MoE.

The middle plot in Figure 2 summarizes the result of conditional coherence. Our method outperforms all the baseline methods except when only one view is given. MMVAE and mmJSD fail to leverage additional input views, showing the performance staying flat or even degrading, due to the incapability of aggregating information inherent in MoE as we discussed in Section 2.4. In contrast, our method monotonically improves the coherence accuracy with more input views, implying that the view-specific encoders are well calibrated by conditional VIBs so that the joint representation aggregated by PoE produces accurate encoding of the digit identity.

The right plot in Figure 2 shows the result of FID scores. Our method achieves the best performance in any combination of input views, which indicates that our method generates more diverse samples in better quality. Figure 3 presents qualitative results of conditional generation from baseline methods and ours. In each row, images above the green line are input observations (views {2} and {2,3} for top and bottom rows respectively) in the test set, whereas images below the line are generated images in view 1. The result shows that our method is successful in cross-view association, especially being much better than any comparing methods at (1) improving the identification of the shared factors using any additional views and (2) expressing the view-specific factors in the target view. Providing more views even improves the results of our method, which can be found in Section B.2.1. Compared to ours, MMVAE, mmJSD, and MoPoE hardly express view-specific diversity in the target view. We hypothesize that their joint representation encoders such as MoE or MoPoE are not sharp enough to discover all the view-specific factors of variation correctly. In contrast, although MVAE uses PoE joint encoder, it poorly preserves the shared factors due to the miscalibration of view-specific encoders. Considering that our method also uses PoE encoder, the performance gap between ours and MVAE shows the effectiveness of conditional VIBs.

### 4.1.2 Multi-View Translation on Caltech-101

We also evaluate our method on the multi-view dataset used in [24] where six visual features are extracted from images in Caltech-101 dataset. In this dataset, each image is associated with six features, which are of Gabor filter [29], Wavelet Moments (WM) [26], CENTRIST [45], Histogram of Oriented Gradients (HOG) [8], GIST [29], and Local Binary Pattern (LBP) [28]. We treat each feature as a view, varying from 40 up to 1984 dimensions. We remark that each feature can be considered as a lossy compression of the original image, where the extracted information from one view may not be necessarily inferrable for other views, thus making it nontrivial to learn the correlation across views.

**Evaluation protocol**  Given the partial observations $\tilde{o}$ at test time, we extract the joint representation $z$ by Eq. (9), and reconstruct the missing view $o_v \notin \tilde{o}$ using the output of the decoder $q_\theta^v(o_v|z)$

(a) Result of training with complete observations    (b) Result of training with partial observations

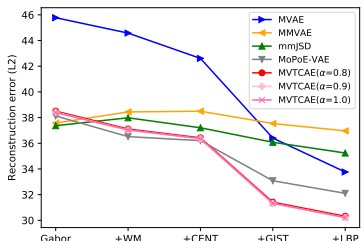 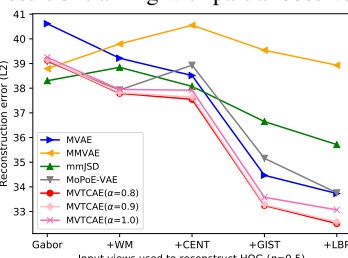

Figure 4: The multi-view translation performance of training with the complete and partial observations, measuring the reconstruction error of the HOG feature by incrementally adding features.

dedicated to view $v$. We then evaluate the L2 reconstruction error. We investigate the impact of missing views on the completeness of the representation. We train our model with complete observations, and apply the model to reconstruct the HOG feature[2] missing in the test time using partial observations $\tilde{o} \in \vec{o}_{\backslash \text{HOG}}$. The performance is averaged over the results of 10 independent runs. We compare to the same baseline methods employed in Section 4.1.

**Results** Figure 4a shows the impact of missing views at test time. We observe that our method effectively decreases the reconstruction error as the input views accumulates in the observation, only with slightly higher error in reconstruction with one view. Interestingly, we notice that the reconstruction error reduces significantly at some point (*i.e.*, when adding GIST feature to the observation). It implies that some views contain more information than the others, and our method is able to learn to utilize this property. On the other hand, MoPoE-VAE is less effective in utilizing GIST or LBP features, while mmJSD and MMVAE hardly show monotonic performance improvement. Remarkably, MVAE performs worst, although the joint encoder model is same as ours, i.e. PoE. As an ablation study, we explore the impact of weighting parameter $\alpha$, which trades off between VIB and conditional VIB (Eq. (8)) in Section B.1.2 in the supplementary material.

## 4.2 Partial Multi-View Representation Learning

To verify that our method can be effectively generalized to the partial observation setting, we conduct experiments on two tasks: multi-view *translation* and *classification*. In both tasks, we aim to show that the representation learned by our method is complete enough to infer the missing views given the incomplete observations (translation task) and useful for downstream tasks (classification task). To simulate the partial observations in both tasks, we follow the protocol of Zhang et al. [50] and generate random view-missing patterns with missing rate $\eta = \sum_{v=1}^{V} U_v/(V \times S)$ ($0 \leq \eta < 1$), where $S$ is the number of entire samples and $U_v$ is the number of samples missing in the $v$-th view.

### 4.2.1 Partial Multi-View Translation on Caltech-101

In this experiment, we investigate if our model can infer complete representations from partial observations. To quantify the completeness of the representation, we employ the multi-view translation task. The goal is to reconstruct a missing view using the representation extracted from other view(s). In this case, the reconstruction error serves as a direct measurement of the completeness.

**Evaluation protocol** Following the same procedures of Section 4.1.2, we repeat the same experiments but using the model trained with incomplete observation ($\eta = 0.5$). This experiment can demonstrate the robustness of our model to missing views in both inference and training time.

**Results** Figure 4b summarizes the experimental results using the incomplete training data ($\eta = 0.5$). We observe that our method exhibits similar trends with complete observation training data, exhibited by monotonic performance improvement in the number of available views, while other methods get negatively affected by additional views when trained with incomplete data (MMVAE, mmJSD, MoPoE-VAE). Interestingly, we observe that MVAE performs better when trained with incomplete data. This is because the missing views in the training data serve as a regularization similar to Dropout [32] or sub-sampled training paradigm [46]. Although MVAE shows monotonic improvement as ours, its performance is turned out to be not comparable to ours.

We also note that our method achieves the best performance with $\alpha = 0.8$ when trained with incomplete data, while the best is achieved at $\alpha = 0.9$ when trained with complete data. We suspect

---

[2]We choose the HOG since it has the largest dimension thus the reconstruction is most nontrivial.

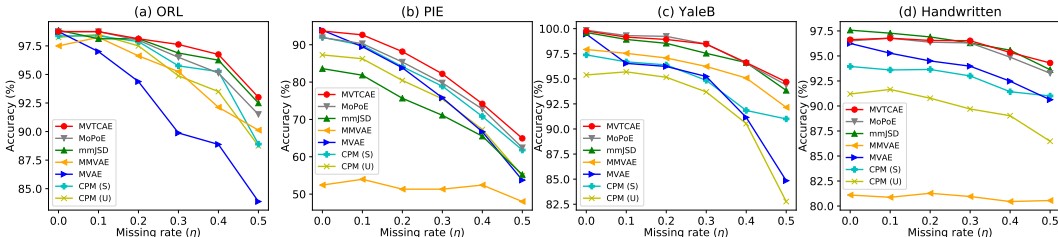

Figure 5: The partial multi-view classification performance under various view missing rate.

that this is because optimizing conditional VIB with partial observation becomes more difficult than with complete observation, and the model tends to solve it via memorization. In this case, the VIB can be useful to improve the generalization. To summarize, our method achieves the best performance for intermediate $0 < \alpha < 1$, which implies that simultaneous optimization of VIB and conditional VIBs in Eq. (8) is effective to solve MVRL and PMVRL. Qualitative results can be found in Section B.2.2 in the supplementary material.

### 4.2.2  Partial Multi-View Classification on Six Datasets [50]

The previous section suggests that our method is able to learn to calibrate representations across views, even when the training data is composed of incomplete observations. In this section, we evaluate the effectiveness of our approach as an unsupervised pre-training of the multi-view representation, investigating if the learned representation is useful for downstream tasks such as classification.

**Datasets**   We evaluate our method on six feature-based image classification datasets used in multi-view learning [50], which are **ORL**,**PIE**, **YaleB**, **CUB**, **Animal** and **Handwritten**. Each dataset is associated with 2 and up to 6 visual features. Similar to the previous section, we treat each feature as a view of data. For all datasets, we follow the same preprocessing and training/test splits used in [50]. See Section D in the supplementary material for a comprehensive description of the datasets.

**Evaluation protocol**   We follow [50] to evaluate our model trained with partial observations with various missing rates $\eta = \{0, 0.1, 0.2, 0.3, 0.4, 0.5\}$. Similar to the experiment in Section 4.1, we first train our model in an unsupervised manner using the observable views. Then we fix the encoders, and train the classifier with labels using the learned representation. To isolate the impact of the additional classifier, we employ the simplest classifier, i.e. logistic regression with the learned representation as input. We report the performance by averaging the results from 10 independent runs.

**Result**   Figure 5 summarizes the results under varying $\eta$ on datasets ORL, PIE, YaleB, and Handwritten, which are with at least three views and thus in our primary interest. Due to the space limit, we present the results on CUB and Animal in Section B.1.3 in the supplementary material. In addition to the baseline methods compared in previous sections, we include two strong baselines, CPM-Nets(S) and CPM-Nets(U), which address the PMVRL with and without label information in the representation learning stage, respectively. Compared to the supervised baseline (CPM-Nets(S)), our method is clearly outperforming in all datasets, even though our model is trained in a purely unsupervised manner. Compared to the unsupervised baseline methods (CPM-Nets(U), MVAE, MMVAE, mmJSD, and MoPoE), our method achieves noticeable improvements, especially when the missing rate is reasonably high ($0.3 \leq \eta \leq 0.5$). It shows that our method is robust in learning the cross-view correlation under partial observations, and the learned representation is informative enough to be useful in downstream tasks, even though we do not use label information in the data or adopt a sophisticated aggregation model in the joint encoder.

## 5  Conclusion

We presented an information theoretic model for unsupervised partial multi-view representation learning. Based on Total Correlation (TC), we derived a novel variational lower bound that allows us to train the model that encodes complete latent representation from partial-view observations. Strictly trained in an unsupervised manner, we also demonstrated that the learned representation is highly effective in downstream tasks, such as multi-view classification and multi-view translation. Although we demonstrated that our method can even learn from partial multi-view data, it still has room for improvement such as learning from unaligned view data, and investigating more sophisticated representation aggregation models for the joint encoder, which we leave as future work.

## Acknowledgments and Disclosure of Funding

This work was supported by the National Research Foundation (NRF) of Korea (NRF-2019R1A2C1087634 red and NRF-2021R1C1C1012540), the Ministry of Science and Information communication Technology (MSIT) of Korea (IITP No. 2019-0-00075, IITP No. 2020-0-00940, IITP No. 2017-0-01779 (XAI), IITP No. 2021-0-00537, and IITP No. 2021-0-02068), the ETRI (Contract No. 21ZS1100), and Samsung Electronics.

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
