# Supplementary Material

## Contents

# A   Theoretical Results

We begin with deriving Eq. (4) in detail since following subsections are based on it.

$$TC_\theta(\vec{O}; Z) = TC(\vec{O}) - TC_\theta(\vec{O}|Z)$$

$$= D_{KL}\left(p_D\left(\vec{o}\right) \| \prod_{i=1}^{V} p_D\left(o_v\right)\right) - \mathbb{E}_{p_\theta(z)}\left[D_{KL}\left(p_\theta\left(\vec{o}|z\right) \| \prod_{i=1}^{V} p_\theta\left(o_v|z\right)\right)\right]$$

$$= \sum_{v=1}^{V} H\left(O_v\right) - H(\vec{O}) - \sum_{v=1}^{V} H_\theta(O_v|Z) + H_\theta(\vec{O}|Z)$$

$$= \sum_{v=1}^{V} H\left(O_v\right) - \sum_{v=1}^{V} H_\theta\left(O_v|Z\right) - H(\vec{O}) + H_\theta(\vec{O}|Z)$$

$$= \sum_{v=1}^{V} I_\theta\left(O_v; Z\right) - I_\theta(\vec{O}; Z), \tag{10}$$

where $p_\theta\left(z\right) = \int p_\theta\left(z|\vec{o}\right) p_D\left(\vec{o}\right) d\vec{o}$, $p_\theta\left(\vec{o}|z\right) = \frac{p_\theta(z|\vec{o}) p_D(\vec{o})}{p_\theta(z)}$, and $p_\theta\left(o_v|z\right) = \int p_\theta\left(\vec{o}|z\right) d\vec{o}_{\backslash v}$ are distributions involved with intractable integration w.r.t. the unknown density $p_D(\vec{o})$.

Thus, we derive three different variational lower bounds on $TC_\theta(\vec{O}; Z)$ introduced in Section 2.1 and Section 2.3 below.

## A.1   Lower Bound that introduces VIB (Eq. (5))

$$TC_\theta(\vec{O}; Z) = \sum_{v=1}^{V} I_\theta\left(O_v; Z\right) - I_\theta(\vec{O}; Z)$$

$$= \sum_{v=1}^{V}\left[\mathbb{E}_{p_\theta(z, o_v)}\left[\ln \frac{p_\theta\left(o_v|z\right)}{p_D(o_v)}\right]\right] - \mathbb{E}_{p_\theta(z, \vec{o})}\left[\ln \frac{p_\theta(z|\vec{o})}{p_\theta(z)}\right]$$

$$= \sum_{v=1}^{V}\left[\mathbb{E}_{p_\theta(z, o_v)}\left[\ln \frac{p_\theta\left(o_v|z\right)}{p_D(o_v)} \cdot \frac{q_\phi^v\left(o_v|z\right)}{q_\phi^v\left(o_v|z\right)}\right]\right] - \mathbb{E}_{p_\theta(z, \vec{o})}\left[\ln \frac{p_\theta(z|\vec{o})}{p_\theta(z)} \cdot \frac{r(z)}{r(z)}\right]$$

$$= \sum_{v=1}^{V}\left[H\left(O_v\right) + \mathbb{E}_{p_\theta(z, o_v)}\left[\ln q_\phi\left(o_v|z\right)\right]\right] - \mathbb{E}_{p_D(\vec{o})}\left[D_{KL}\left[p_\theta(z|\vec{o}) \| r(z)\right]\right]$$

$$+ \sum_{v=1}^{V}\left[\mathbb{E}_{p_\theta(z)}\left[D_{KL}\left[p_\theta\left(o_v|z\right) \| q_\phi\left(o_v|z\right)\right]\right]\right] + D_{KL}\left[p_\theta(z) \| r(z)\right] \tag{11}$$

$$\geq \sum_{v=1}^{V}\left[H\left(O_v\right) + \mathbb{E}_{p_\theta(z, o_v)}\left[\ln q_\phi^v\left(o_v|z\right)\right]\right] - \mathbb{E}_{p_D(\vec{o})}\left[D_{KL}\left[p_\theta(z|\vec{o}) \| r(z)\right]\right]$$

$$= \sum_{v=1}^{V}\left[H\left(O_v\right) + \int\left(\int p_\theta(z|\vec{o}) p_D(\vec{o}) d\vec{o}_{\backslash v}\right) \ln q_\phi^v\left(o_v|z\right) do_v dz\right] - \mathbb{E}_{p_D(\vec{o})}\left[D_{KL}\left[p_\theta(z|\vec{o}) \| r(z)\right]\right]$$

$$= \sum_{v=1}^{V}\left[H\left(O_v\right) + \int p_\theta(z|\vec{o}) p_D(\vec{o}) \ln q_\phi^v\left(o_v|z\right) d\vec{o} dz\right] - \mathbb{E}_{p_D(\vec{o})}\left[D_{KL}\left[p_\theta(z|\vec{o}) \| r(z)\right]\right]$$

$$= \sum_{v=1}^{V}\left[H\left(O_v\right) + \mathbb{E}_{p_\theta(z|\vec{o}) p_D(\vec{o})}\left[\ln q_\phi^v\left(o_v|z\right)\right]\right] - \mathbb{E}_{p_D(\vec{o})}\left[D_{KL}\left[p_\theta(z|\vec{o}) \| r(z)\right]\right], \tag{12}$$

where Eq. (11) is the gap between $TC_\theta(\vec{O}; Z)$ and Eq. (12) (or Eq. (5)). Since $TC_\theta(\vec{O}; Z)$ is upper bounded by $TC(\vec{O})$ which is a constant, maximization of Eq. (12) not only maximizes the original objective $TC_\theta(\vec{O}; Z)$ but also minimizes Eq. (11), the gap between $TC_\theta(\vec{O}; Z)$ and Eq. (12). This results in fitting $q_\phi^v\left(o_v|z\right) \approx p_\theta\left(o_v|z\right)$ and $r(z) \approx p_\theta(z)$[3].

---

[3]In practice, we fix $r\left(z\right) = N(0, \mathrm{I})$ for simplicity.

## A.2 Lower Bound that introduces Conditional VIBs (Eq. (7))

$$TC_\theta(\vec{O}; Z) = \sum_{v=1}^{V} I_\theta(O_v; Z) - I_\theta(\vec{O}; Z)$$

$$= \sum_{v=1}^{V} \left[ \frac{V-1}{V} I_\theta(O_v; Z) + \frac{1}{V} I_\theta(O_v; Z) - \frac{1}{V} I_\theta(\vec{O}; Z) \right]$$

$$= \frac{1}{V} \sum_{v=1}^{V} \left[ (V-1) I_\theta(O_v; Z) - I_\theta(\vec{O}_{\setminus v}; Z \mid O_v) \right] \tag{13}$$

$$= \frac{1}{V} \sum_{v=1}^{V} \left[ (V-1) \mathbb{E}_{p_\theta(z,o_v)} \left[ \ln \frac{p_\theta(o_v|z)}{p_D(o_v)} \right] - \mathbb{E}_{p_\theta(z,\vec{o})} \left[ \ln \frac{p_\theta(z|\vec{o})}{p_\theta(z|o_v)} \right] \right]$$

$$= \frac{1}{V} \sum_{v=1}^{V} \left[ (V-1) \mathbb{E}_{p_\theta(z,o_v)} \left[ \ln \frac{p_\theta(o_v|z)}{p_D(o_v)} \cdot \frac{q_\phi^v(o_v|z)}{q_\phi^v(o_v|z)} \right] - \mathbb{E}_{p_\theta(z,\vec{o})} \left[ \ln \frac{p_\theta(z|\vec{o})}{p_\theta(z|o_v)} \cdot \frac{r_\psi^v(z|o_v)}{r_\psi^v(z|o_v)} \right] \right]$$

$$= \frac{V-1}{V} \sum_{v=1}^{V} \left[ H(O_v) + \mathbb{E}_{p_\theta(z,o_v)} \left[ \ln q_\phi^v(o_v|z) \right] \right] - \frac{1}{V} \sum_{v=1}^{V} \mathbb{E}_{p_D(\vec{o})} \left[ D_{KL} \left[ p_\theta(z|\vec{o}) \| r_\psi^v(z|o_v) \right] \right]$$

$$+ \frac{V-1}{V} \sum_{v=1}^{V} \left[ \mathbb{E}_{p_\theta(z)} \left[ D_{KL} \left[ p_\theta(o_v|z) \| q_\phi(o_v|z) \right] \right] \right] + \frac{1}{V} \sum_{v=1}^{V} \left[ \mathbb{E}_{p_D(o_v)} \left[ D_{KL} \left[ p_\theta(z|o_v) \| r_\psi^v(z|o_v) \right] \right] \right]$$

$$\tag{14}$$

$$\geq \frac{V-1}{V} \sum_{v=1}^{V} \left[ H(O_v) + \mathbb{E}_{p_\theta(z,o_v)} \left[ \ln q_\phi^v(o_v|z) \right] \right] - \frac{1}{V} \sum_{v=1}^{V} \mathbb{E}_{p_D(\vec{o})} \left[ D_{KL} \left[ p_\theta(z|\vec{o}) \| r_\psi^v(z|o_v) \right] \right]$$

$$= \frac{V-1}{V} \sum_{v=1}^{V} \left[ H(O_v) + \int \left( \int p_\theta(z|\vec{o}) p_D(\vec{o}) d\vec{o}_{\setminus v} \right) \ln q_\phi^v(o_v|z) \, do_v dz \right]$$

$$- \frac{1}{V} \sum_{v=1}^{V} \mathbb{E}_{p_D(\vec{o})} \left[ D_{KL} \left[ p_\theta(z|\vec{o}) \| r_\psi^v(z|o_v) \right] \right]$$

$$= \frac{V-1}{V} \sum_{v=1}^{V} \left[ H(O_v) + \int p_\theta(z|\vec{o}) p_D(\vec{o}) \ln q_\phi^v(o_v|z) \, d\vec{o} dz \right]$$

$$- \frac{1}{V} \sum_{v=1}^{V} \mathbb{E}_{p_D(\vec{o})} \left[ D_{KL} \left[ p_\theta(z|\vec{o}) \| r_\psi^v(z|o_v) \right] \right]$$

$$= \frac{V-1}{V} \sum_{v=1}^{V} \left[ H(O_v) + \mathbb{E}_{p_\theta(z|\vec{o}) p_D(\vec{o})} \left[ \ln q_\phi^v(o_v|z) \right] \right]$$

$$- \frac{1}{V} \sum_{v=1}^{V} \mathbb{E}_{p_D(\vec{o})} \left[ D_{KL} \left[ p_\theta(z|\vec{o}) \| r_\psi^v(z|o_v) \right] \right], \tag{15}$$

where $p_\theta(z|o_v) = \int p_\theta(z|\vec{o}) p_D(\vec{o}) d\vec{o}_{\setminus v}$ is a distribution that requires intractable integration w.r.t. the unknown density $p_D(\vec{o})$. Note that the equality in Eq. (13) holds due to the chain rule for MI (see Section A.5). Similar to Eq. (12), maximization of Eq. (15) minimizes Eq. (14), the gap between Eq. (13) and Eq. (15). Thus, our variational optimization scheme fits not only $q_\phi^v(o_v|z) \approx p_\theta(o_v|z)$ but also $r_\psi^v(z|o_v) \approx p_\theta(z|o_v)$.

### A.3 Convex Combination (Eq. (8))

$$TC_\theta(\vec{O}; Z) = (1 - \alpha) \left( \sum_{v=1}^{V} I_\theta (O_v; Z) - I_\theta(\vec{O}; Z) \right)$$

$$+ \alpha \left( \frac{1}{V} \sum_{v=1}^{V} \left[ (V - 1) I_\theta (O_v; Z) - I_\theta(\vec{O}_{\backslash v}; Z \mid O_v) \right] \right)$$

$$= \frac{V (1 - \alpha) + \alpha (V - 1)}{V} \sum_{v=1}^{V} I_\theta (O_v; Z) - \frac{\alpha}{V} \sum_{v=1}^{V} I_\theta(\vec{O}_{\backslash v}; Z \mid O_v) - (1 - \alpha) I_\theta(\vec{O}; Z)$$

$$\geq \frac{V - \alpha}{V} \sum_{v=1}^{V} \left[ H (O_v) + \mathbb{E}_{p_\theta(z|\vec{o}) p_D(\vec{o})} \left[ \ln q_\phi^v (o_v|z) \right] \right]$$

$$- \frac{\alpha}{V} \sum_{v=1}^{V} \mathbb{E}_{p_D(\vec{o})} \left[ D_{KL} \left[ p_\theta(z|\vec{o}) \| r_\psi^v(z|o_v) \right] \right] - (1 - \alpha) \mathbb{E}_{p_D(\vec{o})} \left[ D_{KL} \left[ p_\theta(z|\vec{o}) \| r(z) \right] \right], \quad (16)$$

where $0 \leq \alpha \leq 1$.

### A.4 Interaction Information and its Equivalence to $TC_\theta(\vec{O}; Z)$ when $V = 2$

When there are 2 views, Interaction Information (II) among $O_1$, $O_2$, and $Z$ is defined as follows:

$$I_\theta (O_1; O_2; Z) = I_\theta (O_1; Z) - I_\theta (O_1; Z \mid O_2)$$
$$= I_\theta (O_2; Z) - I_\theta (O_2; Z \mid O_1)$$
$$= I (O_1; O_2) - I_\theta (O_1; O_2 \mid Z)$$

Applying the chain rule of MI (see Section A.5) to the first equality,
we can easily show the equivalence between $I_\theta (O_1; O_2; Z)$ and $TC_\theta(\vec{O}; Z)$:

$$I_\theta (O_1; O_2; Z) = I_\theta (O_1; Z) - I_\theta (O_1; Z \mid O_2)$$
$$= I_\theta (O_1; Z) - (-I_\theta (O_2; Z) + I_\theta (O_1, O_2; Z)) = TC_\theta(\vec{O}; Z) \quad (17)$$

### A.5 Chain Rule for Mutual Information

$$I_\theta (O_v; Z) - I_\theta(\vec{O}; Z) = \mathbb{E}_{p_\theta(z, o_v)} \left[ \ln \frac{p_\theta (z|o_v)}{p_\theta(z)} \right] - \mathbb{E}_{p_\theta(z, \vec{o})} \left[ \ln \frac{p_\theta(z|\vec{o})}{p_\theta(z)} \right]$$

$$= \int \left( \int p_\theta(z|\vec{o}) p_D(\vec{o}) d\vec{o}_{\backslash v} \right) \ln \frac{p_\theta (z|o_v)}{p_\theta(z)} do_v dz$$

$$- \int p_\theta(z|\vec{o}) p_D (\vec{o}) \ln \frac{p_\theta(z|\vec{o})}{p_\theta(z)} d\vec{o} dz$$

$$= \int p_\theta(z|\vec{o}) p_D (\vec{o}) \left( \ln \frac{p_\theta (z|o_v)}{p_\theta(z)} - \ln \frac{p_\theta(z|\vec{o})}{p_\theta(z)} \right) d\vec{o} dz$$

$$= - \int p_\theta(z|\vec{o}) p_D (\vec{o}) \ln \frac{p_\theta(z|\vec{o})}{p_\theta(z|o_v)} d\vec{o} dz = -\mathbb{E}_{p_\theta(z, \vec{o})} \left[ \ln \frac{p_\theta(z|\vec{o}_{\backslash v}, o_v)}{p_\theta(z|o_v)} \right]$$

$$= -I_\theta(\vec{O}_{\backslash v}; Z \mid O_v) \quad (18)$$

## A.6 Connection to Multi-View Information Bottleneck (MIB)

MIB [11] is proposed for learning view-invariant representation between two views. Although one can try to apply MIB to MVRL with more than 2 views by treating it as $\binom{V}{2}$ pair-wise representation learning, it combinatorially scales to the number of given views, making it infeasible to run with many views.

Interestingly, we observe that designing $p_\theta(z, \vec{o})$ as MoE of $r_\psi^v(z|o_v)$ relates conditional VIBs in Eq. (7) to the regularization terms used in MIB for discarding any view-specific information[4].

$$\sum_{v=1}^{V} \mathbb{E}_{p_D(\vec{o})} \left[ D_{KL} \left[ p_\theta(z|\vec{o}) \| r_\psi^v(z|o_v) \right] \right] \tag{19}$$

$$= \sum_{v=1}^{V} \mathbb{E}_{p_D(\vec{o})} \left[ D_{KL} \left[ p_\theta(z|\vec{o}) \| r_\psi^v(z|o_v) \right] \right] + \sum_{i=1}^{V} \mathbb{E}_{p_D(\vec{o})} \left[ D_{KL} \left[ r_\psi^i(z|o_i) \| p_\theta(z|\vec{o}) \right] \right]$$

$$- \sum_{i=1}^{V} \mathbb{E}_{p_D(\vec{o})} \left[ D_{KL} \left[ r_\psi^i(z|o_i) \| p_\theta(z|\vec{o}) \right] \right] \tag{20}$$

$$= \sum_{v=1}^{V} \mathbb{E}_{p_D(\vec{o})} \left[ \int \left( \frac{1}{V} \sum_{i=1}^{V} r_\psi^i(z|o_i) \right) \ln \frac{p_\theta(z|\vec{o})}{r_\psi^v(z|o_v)} dz \right] + \sum_{v=1}^{V} \sum_{i=1}^{V} \left[ \frac{1}{V} D_{KL} \left[ r_\psi^i(z|o_i) \| p_\theta(z|\vec{o}) \right] \right]$$

$$- \sum_{i=1}^{V} \mathbb{E}_{p_D(\vec{o})} \left[ D_{KL} \left[ r_\psi^i(z|o_i) \| p_\theta(z|\vec{o}) \right] \right]$$

$$= \sum_{v=1}^{V} \sum_{i=1}^{V} \mathbb{E}_{p_D(\vec{o})} \left[ \frac{1}{V} \int r_\psi^i(z|o_i) \ln \frac{p_\theta(z|\vec{o})}{r_\psi^v(z|o_v)} dz + \frac{1}{V} \int r_\psi^i(z|o_i) \ln \frac{r_\psi^i(z|o_i)}{p_\theta(z|\vec{o})} dz \right]$$

$$- \sum_{i=1}^{V} \mathbb{E}_{p_D(\vec{o})} \left[ D_{KL} \left[ r_\psi^i(z|o_i) \| p_\theta(z|\vec{o}) \right] \right]$$

$$= \sum_{v=1}^{V} \sum_{i=1}^{V} \mathbb{E}_{p_D(\vec{o})} \left[ \frac{1}{V} \int r_\psi^i(z|o_i) \ln \frac{r_\psi^i(z|o_i)}{r_\psi^v(z|o_v)} dz \right] - \sum_{i=1}^{V} \mathbb{E}_{p_D(\vec{o})} \left[ D_{KL} \left[ r_\psi^i(z|o_i) \| p_\theta(z|\vec{o}) \right] \right]$$

$$\leq \sum_{v=1}^{V} \sum_{i=1}^{V} \mathbb{E}_{p_D(\vec{o})} \left[ \frac{1}{V} D_{KL} \left[ r_\psi^i(z|o_i) \| r_\psi^v(z|o_v) \right] \right]$$

$$= \sum_{v=1}^{V-1} \sum_{i=v+1}^{V} \mathbb{E}_{p_D(\vec{o})} \left[ \frac{2}{V} D_{SKL} \left[ r_\psi^i(z|o_i) \| r_\psi^v(z|o_v) \right] \right], \tag{21}$$

where $D_{SKL} \left[ r_\psi^i(z|o_i) \| r_\psi^v(z|o_v) \right] = \frac{1}{2} D_{KL} \left[ r_\psi^i(z|o_i) \| r_\psi^v(z|o_v) \right] + \frac{1}{2} D_{KL} \left[ r_\psi^v(z|o_v) \| r_\psi^i(z|o_i) \right]$. Remarkably, each of $D_{SKL}$ terms in Eq. (21) is a regularization term used in MIB to discard any information not shared by two views, which encourages each of view-specific encoder to learn view-invariant representation only. Although Eq. (19) is a lower bound on Eq. (21), the gap Eq. (20) between Eq. (21) and Eq. (19) clearly shows that the optimal solutions of Eq. (21) and Eq. (19) have to be equal to:

$$r_\psi^1(z|o_1) = r_\psi^2(z|o_2) = ... = r_\psi^V(z|o_v)$$

Bearing in mind that our goal is to learn complete representation instead of view-invariant representation, Eq. (21) shows that MoE is not a good choice for the conditional VIBs.

## B Comprehensive Experimental Results

In this section, we provide all the evaluations including any quantitative and qualitative results we possibly missed in the main text due to the space limit.

---

[4]view-specific information is called superfluous information in MIB [11].

## B.1 Quantitative Results

We explicitly specify all the quantitative results we presented in Section 4. Some of additional results are included to make our overall experiments more comprehensive.

### B.1.1 Results in multi-view classification / translation on PolyMNIST

Table 1 and 2 specify the numbers used to plot Figure 2.

| Models | Total input views (Linear Classification) | | | | | Total input views (Coherence) | | | |
|---|---|---|---|---|---|---|---|---|---|
| ($\alpha$) | 1 | 2 | 3 | 4 | 5 | 1 | 2 | 3 | 4 |
| MVAE | 0.70 | 0.84 | 0.91 | 0.93 | 0.95 | 0.29 | 0.42 | 0.48 | 0.52 |
| MMVAE | 0.82 | 0.82 | 0.82 | 0.82 | 0.82 | **0.71** | 0.71 | 0.71 | 0.71 |
| mmJSD | 0.89 | 0.98 | 0.99 | **1.0** | **1.0** | 0.69 | 0.57 | 0.64 | 0.67 |
| MoPoE | 0.82 | 0.90 | 0.93 | 0.94 | 0.95 | 0.63 | 0.75 | 0.79 | 0.81 |
| Ours (5/6) | **0.94** | **0.99** | **1.0** | **1.0** | **1.0** | 0.59 | **0.77** | **0.83** | **0.86** |

Table 1: Comparisons on linaer classification and coherence accuracy. All the results are averaged over 5 independent runs. We omit the standard error which are less than 0.01 in most cases.

| Models | Input view(s) | | | |
|---|---|---|---|---|
| ($\alpha$) | View 2 | Views 2,3 | Views 2,3,4 | Views 2,3,4,5 |
| MVAE | $94.06 \pm 5.20$ | $125.87 \pm 5.97$ | $138.46 \pm 6.29$ | $150.53 \pm 6.58$ |
| MMVAE | $228.86 \pm 13.68$ | $224.37 \pm 14.28$ | $220.76 \pm 13.42$ | $217.31 \pm 11.89$ |
| mmJSD | $194.96 \pm 2.75$ | $214.91 \pm 3.69$ | $218.44 \pm 3.52$ | $221.37 \pm 3.64$ |
| MoPoE | $169.70 \pm 2.60$ | $180.53 \pm 2.11$ | $188.92 \pm 3.00$ | $197.33 \pm 3.56$ |
| Ours (5/6) | $\mathbf{90.32} \pm 1.72$ | $\mathbf{99.44} \pm 1.63$ | $\mathbf{111.64} \pm 1.53$ | $\mathbf{122.51} \pm 1.56$ |

Table 2: Comparisons on FID scores averaged over 5 independent runs.

### B.1.2 Ablation study in partial multi-view translation

To investigate the effect of $\alpha$, we compare the performance of our method applying various settings of $\alpha = \{0.0, 0.7, 0.8, 0.9, 1.0\}$. The result is summarized in Table 3 below. In both cases of using complete ($\eta = 0$) and incomplete (0.5) observations, $\alpha \geq 0.7$ yields significant performance improvement comparing to $\alpha = 0.0$. It clearly shows that the conditional VIB ($\alpha = 1$) is very effective on calibrating the representation across views compared to VIB counterpart without cross-view calibration ($\alpha = 0$). Setting $\alpha = 0.9$ and $\alpha = 0.8$ shows the best performance in each case of $\eta = 0$ and $\eta = 0.5$ respectively, which implies that regularization using VIB can be also effective when observations are sparse.

| Models | Views used to reconstruct HOG ($\eta = 0.0$) | | | | | Views used to reconstruct HOG ($\eta = 0.5$) | | | | |
|---|---|---|---|---|---|---|---|---|---|---|
| ($\alpha$) | Gabor | +WM | +CENT. | +GIST | +LBP | Gabor | +WM | +CENT. | +GIST | +LBP |
| MVAE | 45.78 | 44.58 | 42.61 | 36.41 | 33.76 | 40.61 | 39.21 | 38.51 | 34.47 | 33.73 |
| MMVAE | 37.57 | 38.44 | 38.49 | 37.54 | 36.96 | 38.79 | 39.79 | 40.55 | 39.54 | 38.93 |
| mmJSD | **37.37** | 37.98 | 37.21 | 36.08 | 35.24 | **38.30** | 38.84 | 38.07 | 36.65 | 35.71 |
| MoPoE | 38.13 | **36.52** | **36.20** | 33.08 | 32.10 | 39.08 | 37.93 | 38.93 | 35.16 | 33.76 |
| Ours (0.0) | 51.27 | 45.68 | 42.95 | 36.05 | 33.41 | 40.51 | 39.22 | 38.52 | 34.49 | 33.79 |
| Ours (0.7) | 38.56 | 37.17 | 36.48 | 31.53 | 30.43 | 39.16 | 37.85 | 37.56 | 33.29 | 32.58 |
| Ours (0.8) | 38.50 | 37.11 | 36.42 | 31.41 | 30.32 | 39.13 | **37.79** | **37.55** | **33.24** | **32.51** |
| Ours (0.9) | 38.42 | 37.03 | 36.34 | **31.31** | **30.21** | 39.15 | 37.82 | 37.64 | 33.27 | 32.57 |
| Ours (1.0) | 38.38 | 37.04 | 36.36 | 31.33 | 30.22 | 39.25 | 37.95 | 37.92 | 33.58 | 33.07 |

Table 3: The translation performance trained with the complete dataset ($\eta = 0$, from the second to the sixth columns) and incomplete dataset ($\eta = 0.5$, from the seventh to the last columns). We measure the reconstruction error of the HOG by incrementally adding features, accumulated from the feature in the second and seventh columns. The results are the average performance of 10 independent runs. We omit the standard errors which are around 0.06 in most cases.

### B.1.3 Results in partial multi-view classification on 6 datasets including CUB and Animal

In addition to ORL, PIE, YaleB, and Handwritten, Figure 6 shows the partial multi-view classification results on CUB and Animal which are datatsets composed of 2 views. The result shows that our method achieves performance competitive to the strong baseline methods on those 2-view datasets.

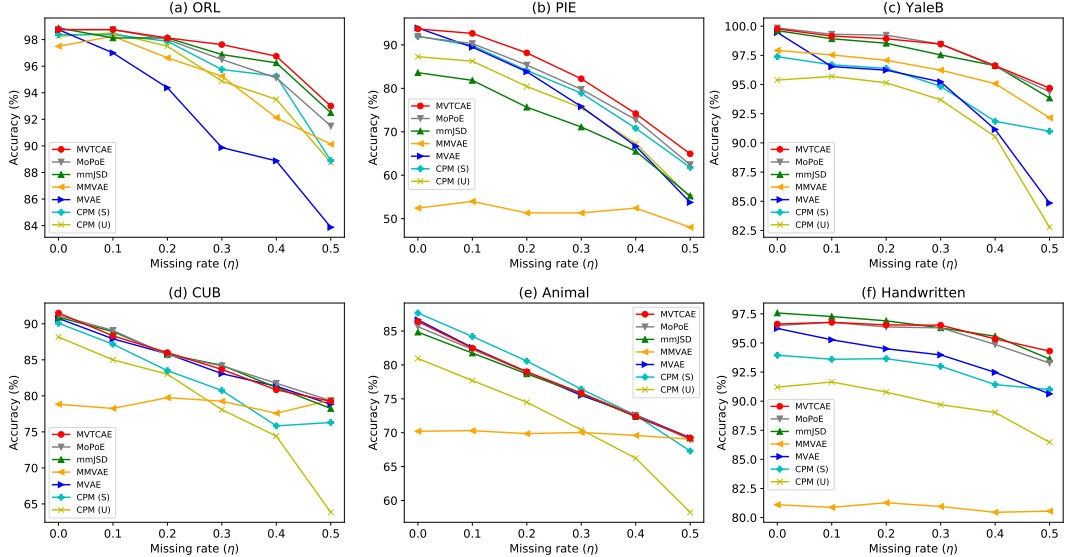

Figure 6: Classification performance on 6 datasets under various view missing rate.

### B.1.4 Results in partial multi-view classification with additional baseline methods

We compare our method using incomplete dataset ($\eta = 0.5$) with additional state-of-the-art MVRL methods such as CCA [18], KCCA [1], DCCA [3], DCCAE [41], DMF-MVC [53], MDcR [48], ITML [9], LMNN [43], and CPM-Nets [50], as well as a naive baseline of concatenating all views (FeatCon). Table 4 summarizes the result. Please note that $\alpha$ is chosen in Figure 5 and Figure 6 according to the result in Table 4 and fixed across all settings of $\eta = \{0, 0.1, 0.2, 0.3, 0.4, 0.5\}$ for each dataset.

| Models | S/U | Datasets (# of views) | | | | | |
| | | ORL (3) | PIE (3) | YaleB (3) | CUB (2) | Animal (2) | HW (6) |
|---|---|---|---|---|---|---|---|
| CCA | U | 38.1 | 37.4 | 66.2 | 57.1 | 24.1 | 55.3 |
| KCCA | U | 42.4 | 33.8 | 67.8 | 57.6 | 23.4 | 56.7 |
| DCCA | U | 38.3 | 35.8 | 67.1 | 40.8 | 9.4 | 54.4 |
| DCCAE | U | 35.6 | 36.3 | 67.6 | 47.5 | 10.4 | 54.4 |
| DMF | U | 60.1 | 34.3 | 57.5 | 30.3 | 47.0 | 55.8 |
| MDcR | U | 65.1 | 23.1 | 58.0 | 70.0 | 61.7 | 55.4 |
| FeatCon | U | 66.3 | 36.3 | 59.8 | 70.8 | 61.9 | 87.1 |
| ITML | S | 76.3 | 36.6 | 81.2 | 70.2 | 56.0 | 73.1 |
| LMNN | S | 70.0 | 56.4 | 76.6 | 73.8 | 59.6 | 86.1 |
| CPM (w/ class) | S | 88.9 | 61.8 | 91.0 | 76.3 | 67.3 | 91.0 |
| CPM(w/o class) | U | $88.8 \pm 0.9$ | $54.9 \pm 1.0$ | $82.8 \pm 1.3$ | $63.8 \pm 1.3$ | $58.3 \pm 0.2$ | $86.5 \pm 0.9$ |
| MVAE | U | $83.9 \pm 1.5$ | $53.8 \pm 0.9$ | $84.8 \pm 0.6$ | $78.8 \pm 0.8$ | $\mathbf{69.2} \pm 0.3$ | $90.6 \pm 0.5$ |
| MMVAE | U | $90.1 \pm 0.9$ | $48.0 \pm 0.7$ | $92.2 \pm 0.8$ | $\mathbf{79.4} \pm 0.9$ | $69.0 \pm 0.4$ | $80.6 \pm 0.4$ |
| mmJSD | U | $92.5 \pm 1.03$ | $55.2 \pm 0.9$ | $93.8 \pm 0.7$ | $78.3 \pm 1.0$ | $\mathbf{69.2} \pm 0.4$ | $93.6 \pm 0.3$ |
| MoPoE | U | $91.5 \pm 0.6$ | $62.4 \pm 1.0$ | $94.4 \pm 0.5$ | $79.3 \pm 0.6$ | $69.0 \pm 0.3$ | $93.3 \pm 0.2$ |
| Ours ($\alpha = 0.8$) | U | $92.6 \pm 0.7$ | $61.7 \pm 0.9$ | $94.2 \pm 0.5$ | $79.2 \pm 0.6$ | $\mathbf{69.2} \pm 0.3$ | $93.2 \pm 0.3$ |
| Ours ($\alpha = 0.9$) | U | $\mathbf{93.0} \pm 0.7$ | $\mathbf{64.9} \pm 0.9$ | $94.0 \pm 0.6$ | $79.0 \pm 0.6$ | $\mathbf{69.2} \pm 0.3$ | $93.6 \pm 0.3$ |
| Ours ($\alpha = 1.0$) | U | $92.8 \pm 0.9$ | $60.5 \pm 1.1$ | $\mathbf{94.7} \pm 0.7$ | $78.6 \pm 0.7$ | $\mathbf{69.2} \pm 0.3$ | $\mathbf{94.3} \pm 0.4$ |

Table 4: Comparisons on classification accuracy (%) with missing rate $\eta = 0.5$. Each dataset is specified with the number of views inside of the parentheses in the second row. S stands for supervised learning and U stands for unsupervised learning in the second column. All the results are averaged over 10 independent runs.

The results when observations are complete ($\eta = 0$) are presented in Table 5 below.

| Models | Datasets (# of views) | | | | | |
|---|---|---|---|---|---|---|
| | ORL (3) | PIE (3) | YaleB (3) | CUB (2) | Animal (2) | HW (6) |
| CPM (w/ class) | $98.4 \pm 0.4$ | $92.0 \pm 0.7$ | $97.4 \pm 0.5$ | $90.1 \pm 0.7$ | $\mathbf{87.7} \pm 0.1$ | $94.0 \pm 0.4$ |
| CPM (w/o class) | $98.3 \pm 0.3$ | $87.3 \pm 1.7$ | $95.4 \pm 0.7$ | $88.2 \pm 1.1$ | $81.0 \pm 0.2$ | $91.2 \pm 0.4$ |
| MVAE | $98.8 \pm 0.3$ | $93.9 \pm 0.3$ | $99.5 \pm 0.3$ | $90.8 \pm 0.6$ | $86.7 \pm 0.3$ | $96.3 \pm 0.3$ |
| MMVAE | $97.5 \pm 0.4$ | $52.4 \pm 1.0$ | $97.9 \pm 0.4$ | $78.8 \pm 1.2$ | $70.2 \pm 0.4$ | $81.1 \pm 0.6$ |
| mmJSD | $98.9 \pm 0.2$ | $83.6 \pm 0.6$ | $99.6 \pm 0.1$ | $90.9 \pm 0.8$ | $84.8 \pm 0.4$ | $\mathbf{97.6} \pm 0.2$ |
| MoPoE | $98.8 \pm 0.3$ | $91.9 \pm 0.4$ | $\mathbf{99.8} \pm 0.1$ | $91.2 \pm 0.7$ | $85.6 \pm 0.4$ | $96.5 \pm 0.3$ |
| Ours ($\alpha = 0.8$) | $98.9 \pm 0.3$ | $\mathbf{94.9} \pm 0.6$ | $99.7 \pm 0.1$ | $91.5 \pm 0.7$ | $86.4 \pm 0.3$ | $96.7 \pm 0.3$ |
| Ours ($\alpha = 0.9$) | $98.8 \pm 0.3$ | $93.7 \pm 0.4$ | $\mathbf{99.8} \pm 0.2$ | $91.5 \pm 0.7$ | $86.4 \pm 0.3$ | $97.0 \pm 0.3$ |
| Ours ($\alpha = 1.0$) | $98.9 \pm 0.3$ | $90.1 \pm 0.5$ | $\mathbf{99.8} \pm 0.2$ | $\mathbf{91.7} \pm 0.7$ | $86.3 \pm 0.3$ | $96.6 \pm 0.3$ |

Table 5: Comparisons on classification accuracy (%) with missing rate $\eta = 0$. All the results are averaged over 10 independent runs.

## B.2 Qualitative Results

We provide comprehensive qualitative results on PolyMNIST along with some examples in the dataset.

### B.2.1 Translation results on PolyMNIST [34] dataset

| | | | | | Digit Identities | | | | | |
|---|---|---|---|---|---|---|---|---|---|---|
| **View** | 0 | 1 | 2 | 3 | 4 | 5 | 6 | 7 | 8 | 9 |
| 1 | | | | | | | | | | |
| 2 | | | | | | | | | | |
| 3 | | | | | | | | | | |
| 4 | | | | | | | | | | |
| 5 | | | | | | | | | | |

Table 6: Examples of samples in PolyMNIST.

Table 6 shows examples of PolyMNIST dataset, where each row is $0 \sim 9$ images in each view. Note that many images in view 1 are remarkably blurry as follows:

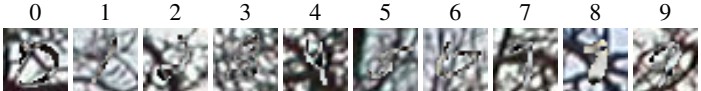

Figures 8, 9, 10, and 11 summarize the qualitative results of conditional generations of each model, where images above the green line are input observations from different view(s) ({2}, {2,3}, {2,3,4}, {2,3,4,5}) in the test set and images below the line are images in view 1 generated by models. Unlike our method, all the baseline methods expose at least one of following three issues:

**Mode collapse in MMVAE, mmJSD, MoPoE**   Generated images in view 1 fail to show diversities in styles of backgrounds and digits, which can be observed by comparing rows in any figures.

**Entangled representations in MMVAE, mmJSD, MoPoE**   Although styles of backgrounds and digits are view-specific factors of variation, comparison among the same columns in Figures 8, 9, 10, 11 shows that those styles get affected by additional observations from new views.

**Discarded shared information in MVAE**   Comparing images above and below the green line in every figure clearly shows the failure in generating coherent samples whose digit identities are supposed to match to conditioned images. Furthermore, it is not obvious that the coherence is improved according to the increased number of given views.

On the other hand, our method expresses view-specific style variations independent of conditioned views while showing better preservation of the digit identities as the number of given views increases.

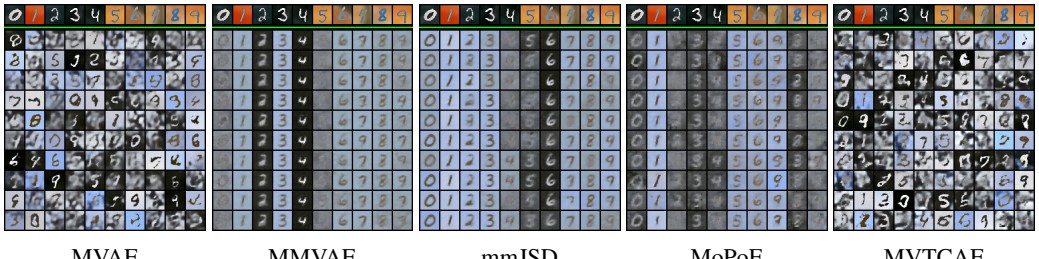

|   MVAE   |   MMVAE   |   mmJSD   |   MoPoE   |   MVTCAE   |

Figure 8: Conditionally generated images of the view 1 given images from the view 2.

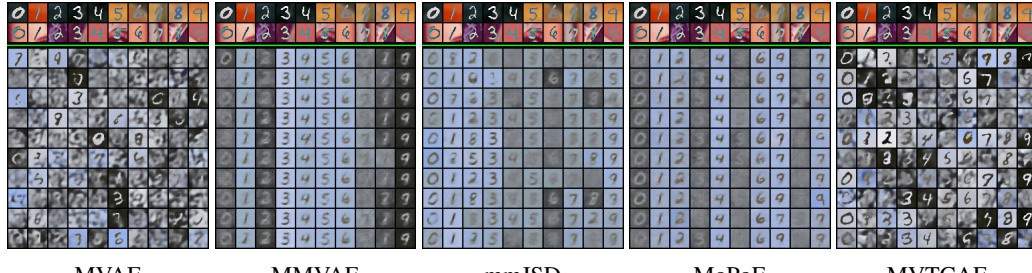

| MVAE | MMVAE | mmJSD | MoPoE | MVTCAE |

Figure 9: Conditionally generated images of the view 1 given images from the views 2 and 3.

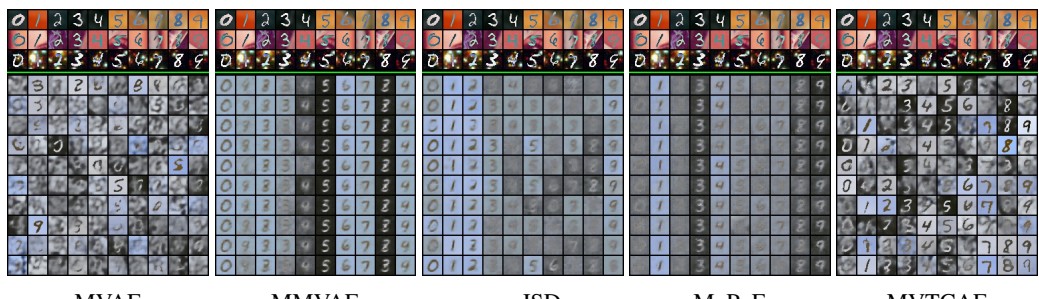

| MVAE | MMVAE | mmJSD | MoPoE | MVTCAE |

Figure 10: Conditionally generated images of the view 1 given images from the views 2, 3, and 4.

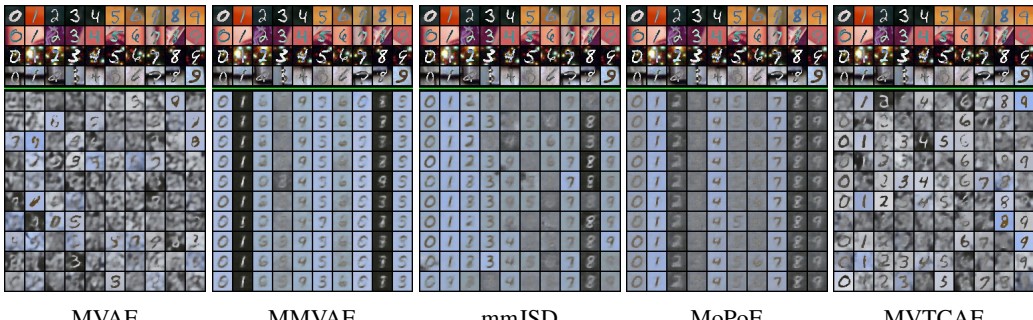

| MVAE | MMVAE | mmJSD | MoPoE | MVTCAE |

Figure 11: Conditionally generated images of the view 1 given images from the rest of views.

We present additional qualitative results in Figures 12, 13, 14, and 15 where images above the green line are conditioned observations from different view(s) ({1}, {1,3}, {1,3,4}, {1,3,4,5}) in the test set and images below the line are images in view 2 generated by models. Three issues we already identified in Figures 8, 9, 10, 11 are similarly observed.

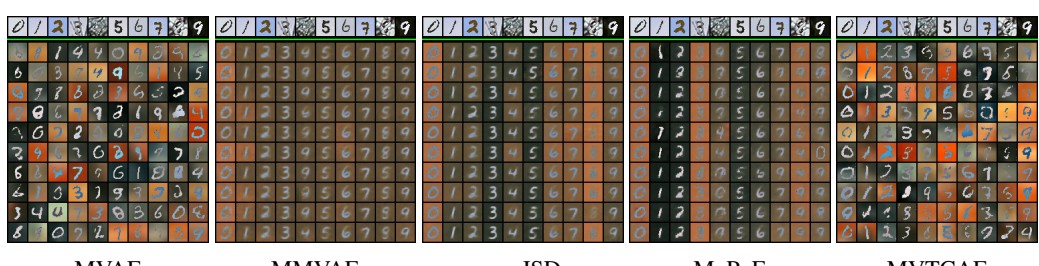

| MVAE | MMVAE | mmJSD | MoPoE | MVTCAE |

Figure 12: Conditionally generated images of the view 2 given images from the view 1.

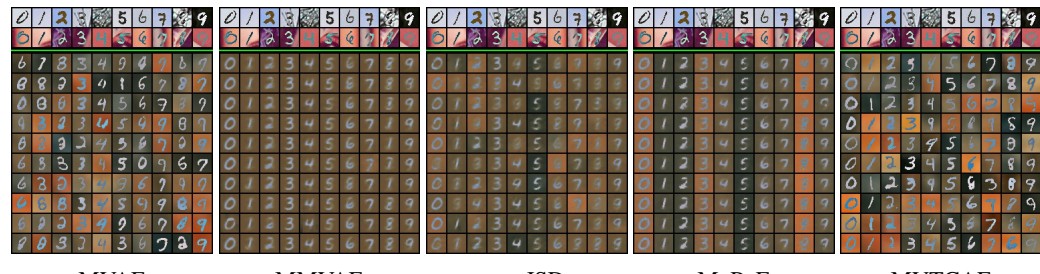

MVAE          MMVAE          mmJSD          MoPoE          MVTCAE

Figure 13: Conditionally generated images of the view 2 given images from the views 1 and 3.

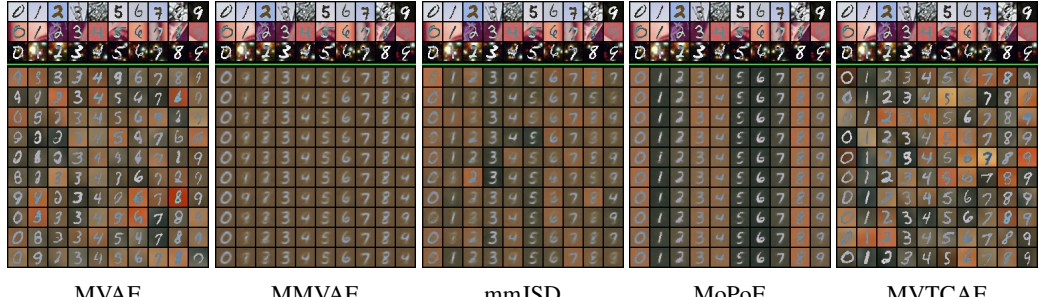

MVAE          MMVAE          mmJSD          MoPoE          MVTCAE

Figure 14: Conditionally generated images of the view 2 given images from the views 1, 3, and 4.

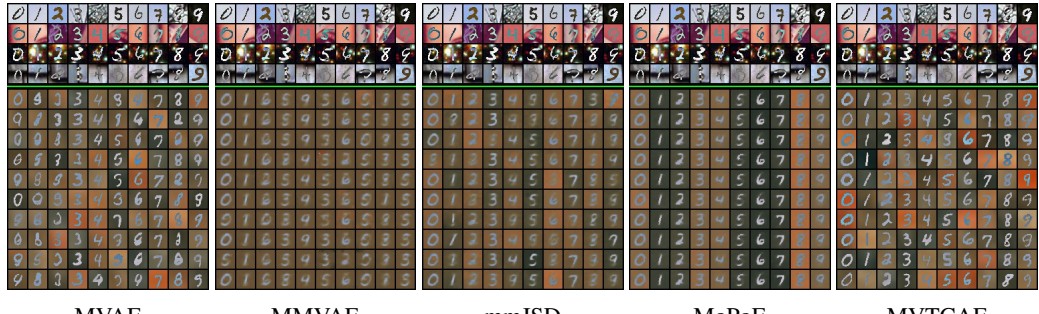

MVAE          MMVAE          mmJSD          MoPoE          MVTCAE

Figure 15: Conditionally generated images of the view 2 given images from the rest of views.

### B.2.2 Translation results in Caltech-101 dataset trained with complete ($\eta = 0$) and incomplete observations ($\eta = 0.5$)

We present qualitative results in Caltech-101 dataset using complete and incomplete training data ($\eta = 0, 0.5$). In Figure 16 and Figure 17, HOG features reconstructed by our model trained with incomplete data ($\eta = 0.5$) show the comparable quality to the ones reconstructed by ours with complete data ($\eta = 0$), demonstrating the robustness of our method to partial observations. In Figure 16 and Figure 17, the labels of features are lamp, starfish, stop sign, motorbike, umbrella, scissors, airplane, butterfly, kangaroo, and watch.

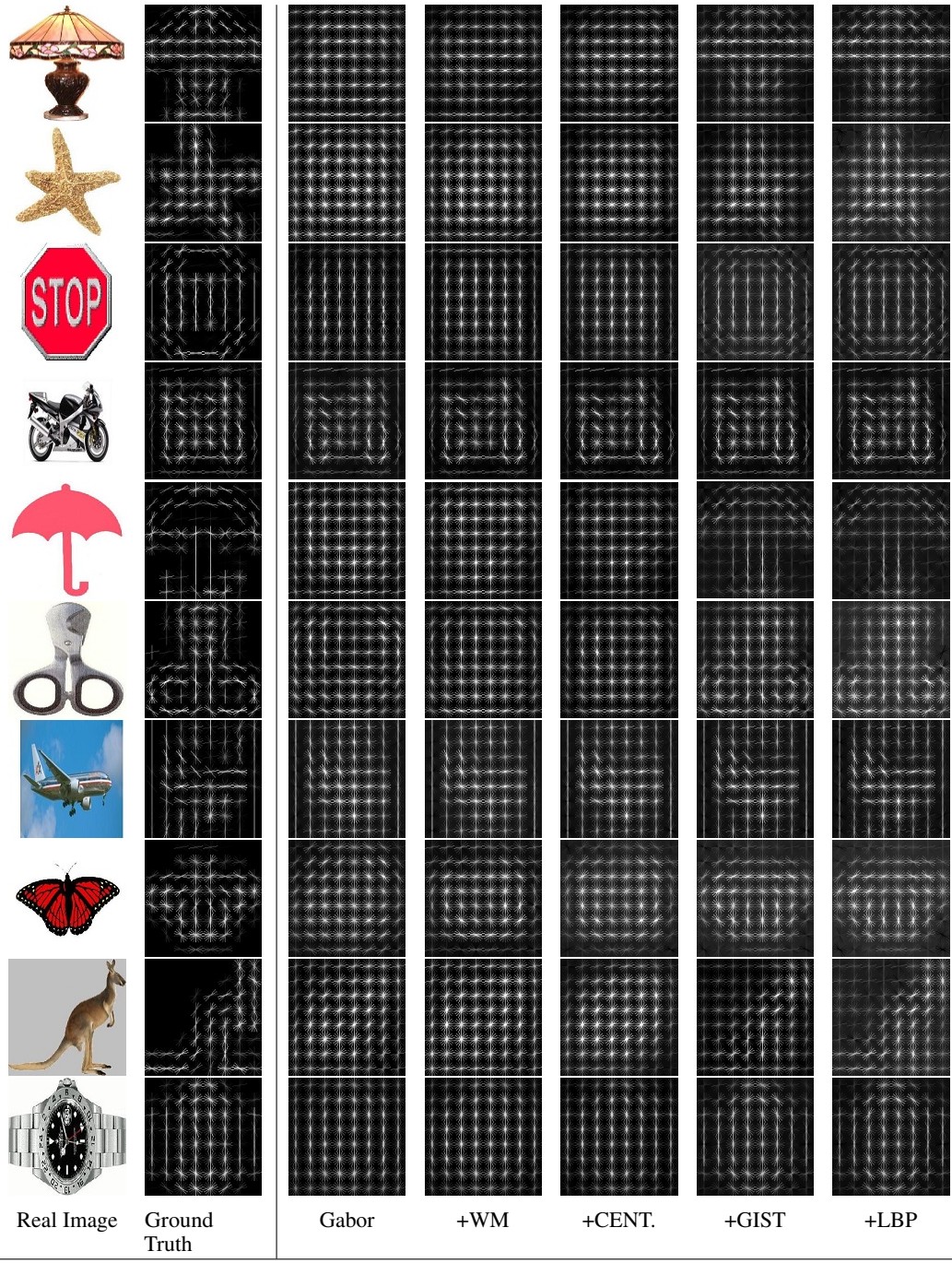

Figure 16: Qualitative results in multi-view translation using complete training data ($\eta = 0.0$). The HOG feature is reconstructed by incrementally adding features, accumulated from the left-most feature (i.e. Gabor).

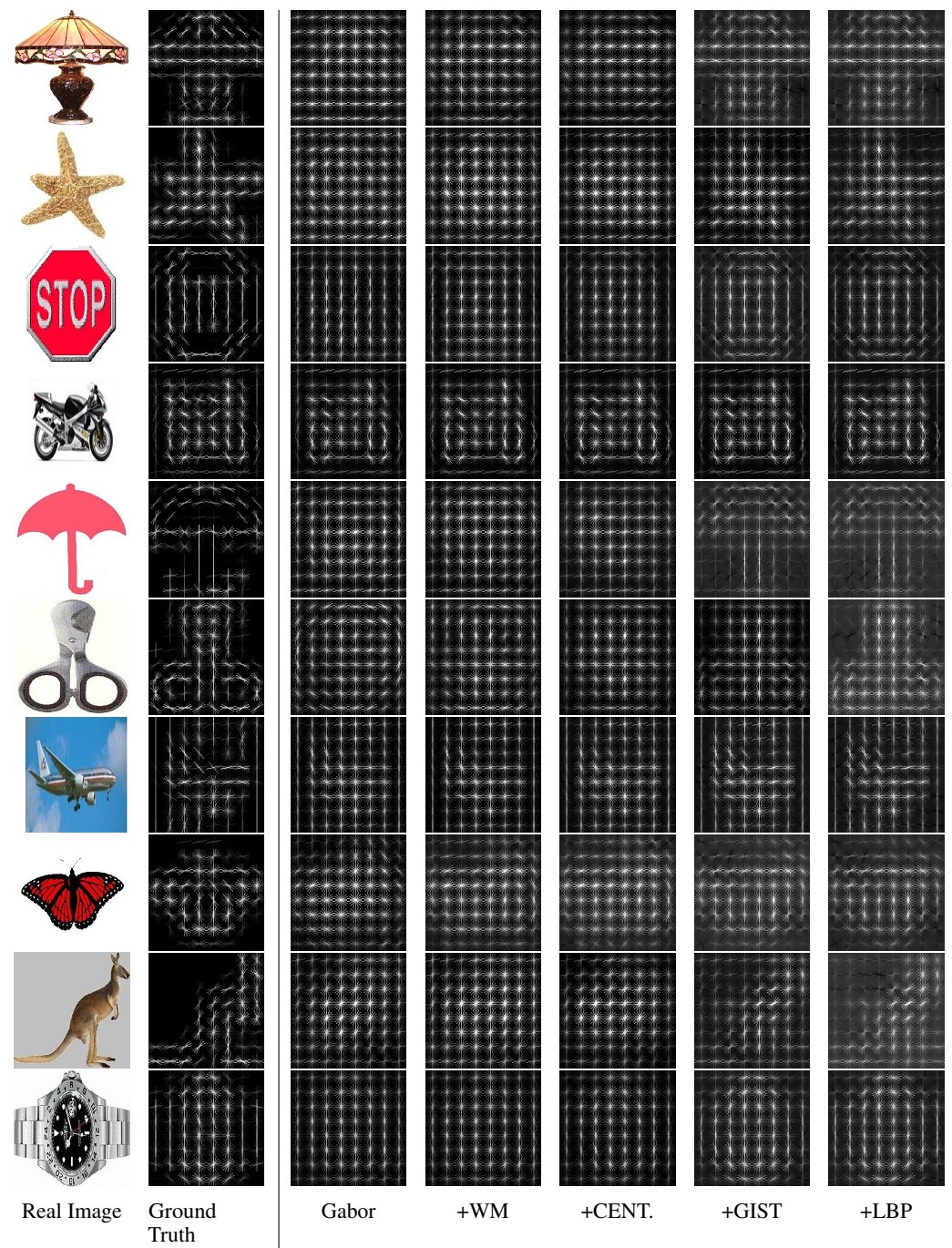

| Real Image | Ground Truth | Gabor | +WM | +CENT. | +GIST | +LBP |
|---|---|---|---|---|---|---|

Figure 17: Qualitative results in multi-view translation using incomplete training data ($\eta = 0.5$). The HOG feature is reconstructed by incrementally adding features, accumulated from the left-most feature (i.e. Gabor).

# C   New Experimental Results on Additional Datasets

To see in depth if our method generalizes well when views are only composed of raw observations, we conducted additional experiment with new datasets, which are Multi-PIE [13] and MNIST-SVHN [30]. Evaluating on those two datasets, we fixed $\alpha = 0.9$ for our method which was found to be reasonable in the previous experiments. Please note that **Multi-PIE** is the source from which **PIE** in Section 4.2.2 composed of 3 hand-crafted features are extracted, and thus they are different.

## C.1   Additional Experimental Results on Multi-PIE in Pixels

To evaluate our method in multiple aspects, we follow the protocol similar to the one used in Section 4.1.1. Specifically, after training all the models in an unsupervised manner using complete observations, we evaluate the learned representation with 4 different metrics, which are linear classification accuracy, conditional coherence accuracy, sample generation quality, and sample diversity. We compare our method with MVAE [46], MMVAE [30], mmJSD [33], and MoPoE-VAE [34] same as Section 4.1.1. For every method, we searched KL coefficient ($\beta$) optimal across all metrics among $\{1, 2.5, 5, 10, 20\}$. Unlike ours and other baseline methods, we were not able to find the optimal $\beta$ for mmJSD that makes the model work commonly well across all tasks. Thus, we report performance of mmJSD with two different settings of $\beta = 1, 20$. Other than $\beta$, we applied same hyperparameters such as epochs, dimensions of latent variable, and batch size to be 300, 128, and 16 respectively/ All the quantitative results below are averaged over 5 seeds (0~4), where as the qualitative results are from the single seed 0.

**Dataset configuration**   Multi-PIE [13] is a dataset composed of 750K bust shot images of 337 human subjects with various facial expressions collected under the circumstance with 15 view points and 19 illumination conditions. Following [35], we extract 250 subjects with 9 poses (within $\pm 60°$), 19 illuminations, and 2 facial expressions and assign the first 200 subjects to training set and the rest 50 for testing set. We group 9 poses into 3 views, where the first view is composed of images with 4 poses within $-60°$, the second view of images in $0°$, and the third view of images in $+60°$. We call first, second, and third views as L (Left), F (Frontal), and R (Right) respectively. For example:

| L (Left) | F (Frontal) | R (Right) |
|:---:|:---:|:---:|

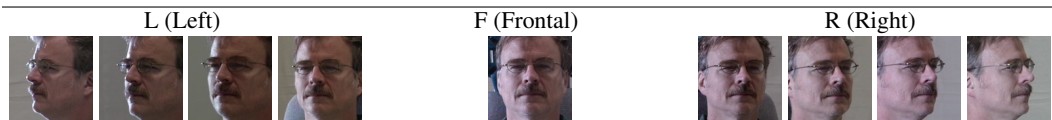

Without applying any view-specific variation in view F, we choose variation in 19 illumination conditions and 2 facial expressions as two shared factors of variation across views while variation in 4 poses in each of views L and R is chosen as a view-specific factor. As a result, each subject owns 38 tuples of 3 images of L,F,R views sharing the illumiation conditions and facial expressions, where the image from each of views L and R is randomly chosen among 4 poses whenever the tuple is sampled.

**Linear Classification**   To apply our method to classification, we fix the encoders and train two linear classifier to predict illumination condition and facial expression using the joint representation extracted from $p_\theta(z|\vec{o})$ feeding complete observations in training set. We count as positive prediction only the case two classifiers simultaneously yields correct predictions on both illumination condition and facial expression. We compute the average classification accuracy over all subsets with the same subset size.

| Model ($\beta$) | Given 1 | Given 2 | Given 3 (All) |
|:---:|:---:|:---:|:---:|
| MVAE (1) | $57.52 \pm 0.83$ | $63.22 \pm 0.53$ | $65.62 \pm 0.65$ |
| MMVAE (1) | $70.5 \pm 0.33$ | $70.61 \pm 0.47$ | $71.14 \pm 0.44$ |
| mmJSD (1) | $75.01 \pm 0.56$ | $77.61 \pm 0.75$ | $78.86 \pm 0.74$ |
| mmJSD (20) | $79.34 \pm 0.66$ | $\mathbf{82.18} \pm 0.76$ | $\mathbf{83.45} \pm 0.93$ |
| MoPoE (1) | $72.55 \pm 0.89$ | $74.03 \pm 0.61$ | $74.23 \pm 0.43$ |
| MVTCAE (10) | $\mathbf{80.48} \pm 0.67$ | $81.87 \pm 0.79$ | $82.07 \pm 0.81$ |

Table 7: Joint classification accuracy of 19 illumination conditions and 2 facial expressions using the learned latent representation.

Table 7 summarizes the result of linear classification accuracy according to the number of input views. The result shows that mmJSD (20) and ours show the best performances whose error bars overlap, which implies that both methods can successfully extract the information shared across views.

**Conditional coherence**   To measure the conditional coherence accuracy, we extract the representation of every subset of views using $p_\theta$ and generate views that are absent in the subset using $q_\phi$. Those generated views are fed into the pretrained CNN-based classifier. We count as a correct prediction if the prediction on both the illumination condition and the facial expression from the classifier simultaneously matches two labels of the input view images. The results are averaged over all subsets with the same size.

| Model ($\beta$) | Target View | | | # of Input Views | |
|---|---|---|---|---|---|
| | L | F | R | Given 1 | Given 2 |
| MVAE (1) | $8.84 \pm 0.69$ | $39.7 \pm 2.86$ | $9.08 \pm 0.29$ | $18.09 \pm 1.15$ | $21.44 \pm 1.15$ |
| MMVAE (1) | $74.66 \pm 0.68$ | $85.1 \pm 0.25$ | $72.95 \pm 0.89$ | $77.54 \pm 0.41$ | $77.62 \pm 0.44$ |
| mmJSD (1) | $69.12 \pm 1.97$ | $83.45 \pm 0.61$ | $65.65 \pm 2.07$ | $73.7 \pm 1.07$ | $70.82 \pm 1.03$ |
| mmJSD (20) | $60.08 \pm 0.47$ | $75.22 \pm 0.34$ | $55.8 \pm 1.39$ | $68.77 \pm 0.62$ | $53.55 \pm 0.55$ |
| MoPoE (1) | $76.17 \pm 0.32$ | $85.37 \pm 0.45$ | $73.04 \pm 0.8$ | $77.36 \pm 0.45$ | $79.85 \pm 0.52$ |
| MVTCAE (10) | $\mathbf{82.58} \pm 0.6$ | $\mathbf{85.81} \pm 0.15$ | $\mathbf{82.77} \pm 0.5$ | $\mathbf{83.02} \pm 0.28$ | $\mathbf{85.1} \pm 0.26$ |

Table 8: Joint coherence accuracy in the conditionally generated samples with respect to illumination conditions and facial expressions.

Table 8 summarizes the results of conditional coherence accuracy in two ways according to the target view and the number of input views. The result shows that our method outperforms all the comparing methods across all aspects. The results indicates that conditional VIBs in our method are very effective to identifying the shared factors of variation and improving preservation of them using additional input views. On the other hand, MVAE shows poor performance incomparable to any comparing methods, which implies that augmenting ELBO of each view to the ELBO of the joint views harms the preservation of shared factors of variation.

**Sample quality**   To evaluate the sample quality of conditional generation, we reuse images generated for evaluating the conditional coherence by comparing them to the ground truth images in the target view paired with their input view images. We quantify similarities between those generated images and corresponding target images using LPIPS [52], which measures perceptual distance between two images. Considering that each of generated images in view L and R can have any of 4 different poses, we compute LPIPS distance between the generated images and each of 4 target images and count the minimum distance. The results are averaged over all subsets of input views with the same size.

| Model ($\beta$) | Target View | | | # of Input Views | |
|---|---|---|---|---|---|
| | L | F | R | Given 1 | Given 2 |
| MVAE (1) | 0.3262 | 0.1807 | 0.3180 | 0.2785 | 0.2679 |
| MMVAE (1) | 0.2953 | 0.1812 | 0.2931 | 0.2565 | 0.2566 |
| mmJSD (1) | 0.3207 | 0.1992 | 0.3180 | 0.2758 | 0.2863 |
| mmJSD (20) | 0.3595 | 0.2346 | 0.3564 | 0.3048 | 0.3409 |
| MoPoE (1) | 0.2868 | 0.1741 | 0.2855 | 0.2499 | 0.2466 |
| MVTCAE (10) | **0.2202** | **0.1673** | **0.2211** | **0.2046** | **0.1995** |

Table 9: LPIPS distance between generated samples and target images. Since views L and R have variation of 4 different poses as their own factors of variation, there are four candidate target images per generated sample if its target view is L or R. Thus, we count the minimum distance out of 4. Standard errors are omitted since they are negligibly small.

Table 9 summarizes the results of the sample quality evaluation in two ways according to the target view and the number of input views. The results shows that our method outperforms all the comparing methods across all aspects. Compared to MMVAE, mmJSD, and MoPoE-VAE, our method shows significant performance gap in the case target view is L or R while the gap is relatively small in the case the target view is F. This is because those methods are using MoE as their joint representation encoder that hardly expresses view-specific factors of variation, which results in generating blurry images collapsing to one pose in views L and R (see Figure 19). Although MVAE generates samples with variation in poses, those samples are not consistent to the given subject.

**Sample diversity** We measure the view-specific diversity in the generated samples by entropy. We extract the representation of subset of views $\{L, F, R, LF, FR\}$ using $p_\theta$ and generate 10 samples (per instance) in any of views L,R that are absent in the input subset using $q_\phi$. Those 10 generated images are fed into another pretrained CNN-based classifier which predicts the pose among 4 candidates in the target view. To compute entropy, we first apply one-hot encoding to 10 predicted labels. Then we normalize those 10 encodings to make their sum to be 1 and compute entropy which stands for the diversity with respect to the pose. The results are averaged over all subsets with the same size.

| Model ($\beta$) | Target View | | # of Input Views | |
| --- | --- | --- | --- | --- |
| | L | R | Given 1 | Given 2 |
| MVAE (1) | $1.65 \pm 0.01$ | $1.59 \pm 0.01$ | $1.68 \pm 0.01$ | $1.48 \pm 0.02$ |
| MMVAE (1) | $0.05 \pm 0.0$ | $0.06 \pm 0.01$ | $0.05 \pm 0.0$ | $0.05 \pm 0.0$ |
| mmJSD (1) | $0.15 \pm 0.02$ | $0.16 \pm 0.01$ | $0.14 \pm 0.01$ | $0.19 \pm 0.01$ |
| mmJSD (20) | $0.42 \pm 0.03$ | $0.42 \pm 0.02$ | $0.37 \pm 0.01$ | $0.52 \pm 0.01$ |
| MoPoE (1) | $0.07 \pm 0.01$ | $0.07 \pm 0.01$ | $0.08 \pm 0.01$ | $0.05 \pm 0.0$ |
| MVTCAE (10) | $\mathbf{1.70} \pm 0.01$ | $\mathbf{1.66} \pm 0.01$ | $\mathbf{1.72} \pm 0.0$ | $\mathbf{1.6} \pm 0.01$ |

Table 10: Diversity of poses in the generated samples.

Table 10 summarizes the result of measuring diversity in the generated samples by their entropy. The result shows that our method significantly outperforms all the MoE-based methods (MMVAE, mmJSD, MoPoE-VAE) due to their issues on preserving view-specific factors as we discussed in Section 4.1.1. Our method even outperforms MVAE, which implies that conditional VIBs in our method are greatly effective to the cross-view association without introducing any side effects.

**Qualitative results** Lastly, Figure 19 presents 4 samples of conditionally generated samples in each of target views L and R feeding an image of the first subject in view F in the test set. The results show that our method (bottom row) generates samples in the finest quality with the best preservation of the subject's identity and the highest diveriisy in pose, which is consistent with what we observed in the quantitative results above.

| GT in **F** | Generated samples in **L** | Generated samples in **R** |
| --- | --- | --- |

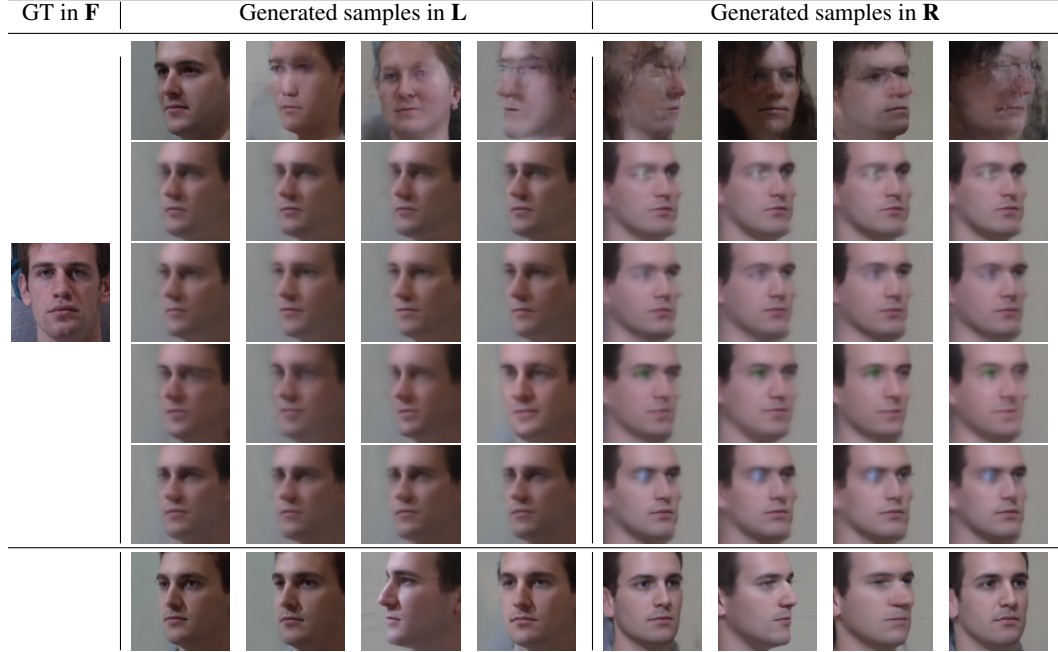

Figure 19: Examples of conditional generation. The first 5 rows are results from MVAE, MMVAE, mmJSD(1), mmJSD(20), and MoPoE. The bottom row is the result of our method.

**Summary** Showing state-of-the-art performance in the task of classification using the learned representation, our algorithm absolutely outperforms all the baseline methods in the translation tasks. It is remarkable that (1) even successfully preserving the information shared across views (observed in Table 7, 8), our method generate samples not only in the best quality (observed in Table 9 and Figure 19) but also in the highest diversity (observed in Table 10 and Figure 19).

## C.2 Additional Experimental Results on MNIST-SVHN

The table below summarizes the result of the experiment in MNIST-SVHN (MS) dataset averaged over 5 runs with seeds $0 \sim 4$. The evaluation protocol, codebase, and hyperparameter settings follow the experiments in the MoPoE-VAE [34] on MNIST-SVHN-Text (MST) dataset, which augments to the original MS the text of digit IDs as the third modality. Please note that all the results below are from the baseline implementations in the MoPoE-VAE codebase whose VAE architectures dedicated to MNIST and SVHN are the same as the ones used by MMVAE. Lastly, we simply discarded the third modality.

| Models | Representation Classification (RC) | | | Coherent Generation (CG) | | |
|--------|-----------|----------|-------|-------|------|------|
| | M (MNIST) | S (SVHN) | MS | Joint | M→S | S→M |
| MVAE | 87.38 | 58.23 | 87.37 | 42.98 | 56.65 | 35.63 |
| MMVAE | 72.83 | 60.89 | 66.89 | 42.45 | 26.76 | **74.94** |
| mmJSD | 88.58 | **81.44** | 93.81 | 12.73 | 22.33 | 65.44 |
| MoPoE | 82.48 | 70.62 | 87.96 | 44.95 | 21.23 | 72.38 |
| MVTCAE | **93.48** | 77.99 | **94.97** | **46.71** | **81.09** | 59.91 |

where representation classification (RC) is measured by the accuracy of the single linear classifier trained on the latent representation as input (z from M only, z from S only, and z from jointly M and S), and coherent generation (CG) is measured by the accuracy of the pretrained CNN classifier whose input is the image generated by each model (e.g. Joint is measured from MNIST and SVHN images generated from the same z sampled from the prior distribution, and M->S is measured by SVHN images generated from MNIST images).

Among 6 different evaluation results, our method outperforms baseline methods in 4 tasks (RC / MNIST and MS, CG / Joint and M->S) and performs competitive to the baselines in 1 task, RC / SVHN. Our method performs relatively poor only in CG / S->M (ranked 4th), the advantage of our method in M->S is much more noticeable. Comparing to strong baseline methods such as MMVAE, mmJSD, and MoPoE-VAE, our method shows more balanced performance in two different directions of CG, as measured in Joint, achieving the best average performance.

# D Dataset Statistics

We present information on all the datasets we used in Section 4 in detail. Each feature is treated as one view in every dataset.

## D.1 In Section 4.1.1

- **PolyMNIST** [34] is an image dataset composed of 5 views each of which is created by fusing each of MNIST images with a background, a 28x28x3 sized patch randomly cropped from one of five different images chosen by [34]. Each MNIST images is binarized, and colors of its background image is inverted at the locations where the digit of the MNIST is inserted. Some examples are showcased in Section B.2.1. License information of those 5 images used for backgrounds can be found in [34].

## D.2 In Section 4.1.2 and 4.2.1

- **Caltech-101** [23] is a image dataset collected for object recognition task. Images in Caltech-101 are categorized as 101 different classes. From Caltech 101, six visual features are extracted and compiled as a multiview dataset by Li et al. [24], which are are 48 dimensional Gabor feature [29], 40 dimensional wavelet moments (WM), 254 dimensional CENTRIST [45] feature, 1984 dimensional HOG [8] feature, 512 dimensional GIST [29] feature, and 928 dimensional LBP [28] feature.

## D.3 In Section 4.2.2

1. **ORL**[5] is a dataset composed of 400 facial images of 40 subjects. 4096 dimensional Intensity feature, 3304 dimensional LBP feature, and 6750 dimensional Gabor feature are extracted.

2. **PIE**[6] consists of 750K bust shot of 337 human subjects. A subset which contains 10 images for each of 68 people is collected, 680 images in total. We use 484 dimensional Intensity feature, 256 dimensional LBP feature, and 279 dimensional Gabor feature extracted from the subset.

3. **Yale Face Database B**[7] (YaleB) is a database which contains 5850 images of 10 subjects captured with 585 different illumination conditions (65 illumination conditions for 9 different poses). A subset which contains 650 images of 10 subjects is collected. We use 2500 dimensional Intensity feature, 3304 dimensional LBP feature, and 6750 dimensional Gabor feature extracted from the subset.

4. **CUB** [40] is a dataset consists of 11788 images of birds that belong to 200 different classes. A subset of 600 images that covers 10 categories are collected. 1024 dimensional GoogLeNet visual feature and 300 dimensional doc2vec feature are are extracted from the subset.

5. **Animal** is a dataset composed of 10158 images of animals distributed across 50 classes. Two different deep visual features are extracted, which are 4096 dimensional DECAF feature and 4096 dimensional VGG19 feature.

6. **Handwritten**[8] is a dataset that contains 2k handwritten digits of 0 to 9. Six features are generated, which are 76 dimensional Fourier coefficients of the character shapes feature, 216 dimensional profile correlations feature, 64 dimensional Karhunen-love coefficients feature, 240 dimensional ($2 \times 3$) pixel averages feature, 47 dimensional Zernike moment feature, and 6 dimensional morphological feature.

Note that subsamples and features of ORL, PIE, YaleB, CUB, and Animal datasets are collected by Zhang et al. [50]. As a result, there are 3 features in ORL, PIE, YaleB and 2 features in CUB, Animal, whereas 6 features in Handwritten. Lastly, we followed the same preprocessing and training/test splits used in Zhang et al. [50] for all six datasets employed in Section 4.2.2.

---

[5]https://www.cl.cam.ac.uk/research/dtg/attarchive/facedatabase.html
[6]http://www.cs.cmu.edu/afs/cs/project/PIE/MultiPie/Multi-Pie/Home.html
[7]http://vision.ucsd.edu/~leekc/ExtYaleDatabase/Yale%20Face%20Database.htm
[8]https://archive.ics.uci.edu/ml/datasets/Multiple+Features

# E Implementation Details

We report implementation details in each experiment including the hyperparameters and the structures of encoders and decoders. We used two codebases, one is implemented in PyTorch by MoPoE-VAE[9] and the other is written in TensorFlow by CPM-Nets[10]. Except the joint representation encoder uniquely determined by each model, MVAE, MMVAE, mmJSD, MoPoE-VAE, and our method share the same network architectures and hyperparameter settings including the batch size, the size of encdoer/decoder and latent variables, coefficient of reconstruction ($w$) and KL regularizations ($\beta$) terms, and epochs. In the test phase, the representation fusion in each model is conducted using its own joint representation encoder (as identified in Section 2.4), except mmJSD which learns to fuse representations using its dynamic prior. For dimensionalities of inputs of encoders and outputs of decoders, please read Section D. Further information can be found in our official implementation[11].

## E.1 In Section 4.1.1

Following MoPoE-VAE, we fixed $w = 1$ and $\beta = 2.5$ and ran for 300 epochs with 5 seeds ($0 \sim 4$). We also set network structures and dimension size of the latent variable (512) same as MoPoE-VAE. We fixed $\alpha$, the only hyperparameter our method uniquely has, to be $\frac{5}{6}$ that equally weights the VIB and conditional VIBs.

## E.2 In Section 4.1.2 and 4.2.1

Fixing $w = 200$ and $\beta = 1.0$, we ran each model with 10 seeds ($0 \sim 9$) for 10,000 epochs to ensure that all the methods are converged. As an ablation study, we evaluated our method with various settings of $\alpha = \{0.0, 0.7, 0.8, 0.9, 1.0\}$ as we reported in Section B.1.2, which results in $\alpha = 0.9$ and $\alpha = 0.8$ showing the best performance when $\eta = 0.0$ and $0.5$ resepectively. We adopted following network architectures with 100-dimensional latent variables for all methods.

| Dataset | Caltech 101 |
|---------|-------------|
| | |
| Network | Encoder $r_\psi^v(z\|o_v)$ |
| Input | $o_v$ |
| Layer 1 | FC. 200. ReLU |
| Layer 2 | $2\times$ FC. 100 ($\mu_v, \log \sigma_v^2$) |
| | |
| Network | Decoder $q_\phi^v(o_v\|z)$ |
| Input | $z \sim p_\theta(z\|\vec{o})$ |
| Layer 1 | FC. 200. ReLU |
| Layer 2 | FC. dim($o_v$) |

## E.3 In Section 4.2.2

We ensured that structures and sizes of our decoders are same as the ones used in the official implementation of CPM-Nets. The only difference is the activation function being used. We used ReLU in the middle of two fully connected (FC) layers. We chose the structures of our view-specific encoders as the reverse of decoders, ensuring that the sizes of the latent variables we use are same as the ones used in CPM-Nets as well. We described how $\alpha$ is chosen in Section B.1.4. For MVAE, MMVAE, mmJSD, MoPoE, and ours, we applied the same encoder/decoder structures and ran for the same number of epochs with 10 seeds ($0 \sim 9$) per dataset to make fair comparison. We chose $w = 100$ and $\beta = 1.0$ for all datasets. Dimensions of the latent variable and the epoch per dataset are specified below.

| Hyperparamters | Datasets | | | | | |
|----------------|------|------|-------|------|--------|-------------|
| | ORL | PIE | YaleB | CUB | Animal | Handwritten |
| Dimensions of $z$ | 256 | 150 | 128 | 128 | 512 | 128 |
| Epochs | 1,000 | 5,000 | 5,000 | 2,000 | 100 | 5,000 |

Table 11: Hyperparameters used in ORL, PIE, YaleB, CUB, Animal, and Handwritten datasets.

---

[9]https://github.com/thomassutter/MoPoE

[10]https://github.com/hanmenghan/CPM_Nets

[11]https://github.com/gr8joo/MVTCAE

# F   Computation Resources

We used 10 systems equipped with following devices.

CPU: Intel(R) Core(TM) i7-9700K CPU @ 3.60GHz

Memory: 32 Gb.

GPU: TITAN Xp

# G   Societal Impact

Positively, our method could be used to reduce the number of sensors in multi-sensor system without losing sensor fusion accuracy, reducing carbon footprint and environmental waste due to redundant sensors. Negatively, we see the possibility that our method could be exploited in wrongful manner, such as Deepfake. Specifically, one might adopt our method to synthesize someone's image in the representation space and generate fake samples for fraudulent purposes. Similarly, our method can be utilized in synthesizing voice for impostors.