# OpenReview forum: "Multi-View Representation Learning via Total Correlation Objective"
_NeurIPS.cc/2021/Conference — NeurIPS 2021 Poster_

### Official Review · Reviewer_yxVq · 2021-07-11

**Rating:** 6
**Confidence:** 4

**Summary:**

This paper proposes a novel Multi-View Representation Learning method to learn a shared representation of multiple observations. The objective is to learn a complete representation through maximizing the reduction of total correlation. The method is based on the VAE framework and is capable of dealing with missing views.

**Limitations And Societal Impact:**

I think the writeup could be improved, especially the order of different modules. Besides, Fig. 4 should be moved to the main text with more descriptions. The whole pipeline would be more clear.

**Main Review:**

*originality*

The paper proposes a novel objective function for Multi-View Representation Learning (MVRL). The objective optimizes a lower bound of the total correlation between the views and the latent representation.

*quality*
- In Eq. (6), you mention "the latent representation is regularized to be inferrable from any view and thus cannot be dominated by a few one". But the MI terms could still be high for some views and low for some other views. You are taking the average in the end anyway. I don't think this solves the imbalance issue.
- In Eq. (7), it seems that both $p(z|o)$ and $r(z|o_v)$ are learnable posteriors. Would this cause any training instability?
- PoE seems to be an aggregation of all the individual posteriors and definitely has a larger support than individual ones. This could also bring some numerical concerns. How did you address this issue?
- When you say "VIB in Eq. (5) favors the minimal sufficient encoding", I'm ok with the minimal due to the minus. But I don't think this term really encourages "sufficient". This term could potentially cause "insufficiency".
- For the KL terms, did you use analytical form or sample estimation?
- In Fig. 1, MMVAE's performance cannot be improved with more views. What might cause this?
- In Fig. 3, there seems to be no significant gap between MVTCVAE and other unsupervised methods (mmJSD, MoPoE). Besides, as the missing rate increases, the performance drop trends are quite similar. Then I wonder if carefully tuning other methods could achieve the same goal.

*clarity*
- Authors should really put Fig. 4 to the main text rather than the appendix.
- When you bring up the limitations "Missing views", as a reader, I thought you would immediately explain how you address this. But I have to wait till the end of section 2.4. I would suggest you explaining section 2.4 first.

*significance*
It is novel to propose such an idea for MVRL. But maybe authors can demonstrate more scenarios besides the accuracies.


**Time Spent Reviewing:**

4

---

> ### Author Response · Authors · 2021-08-10
> **Response to Reviewer yxVq**
>
> We greatly appreciate your thoughtful and detailed feedback.
>
> **1. Resolving imbalanced representation**\
> To see how conditional IB in Eq.(6) encourages learning balanced representation, it is easier to analyze its variational lower-bound (conditional VIB in Eq.(7)). Specifically, we observe that conditional VIB is the average KL distance between representations from the joint encoder $p_{\theta}(z|\vec{o})$ and a view specific encoder $r_{\psi}^{v}(z|o_{v})$, and has its minimum at 0. When the model learns an imbalanced representation, the joint representation is dominated by certain views, and the KL divergence rapidly increases in views that have less contributions in $p_{\theta}(z|\vec{o})$. Since KL divergence has no upper-bound, this can lead to significant penalty in the objective function, and the model is naturally encouraged to learn balanced representation to minimize conditional VIB (i.e., it should learn information in the joint encoder inferable from all view-specific encoders). This is in contrast to the objective of VIB (Eq.(5)), where it simply penalizes the total amount of information in the representation encoded by the joint encoder without ensuring that every view is contributing to the joint representation or is optimized.
>
> **2. Stability of optimizing encoders**\
> We appreciate your comment. Although we did not observe any numerical issues of optimizing PoE encoders in our formulation, which can be verified in the submitted code, we hypothesize that there are two reasons for the stable training as follows:
>
> 1) All the KL terms are optimized with their analytic solutions (since all the encoders are Gaussian), which are computed over the complete support sets.
> 2) The joint encoder distribution $p_{\theta}(z|\vec{o})$ is modeled by the combination of view-specific ones $r^{v}_{\psi}(z|o_v)$ (Eq.(9)), which can be useful to avoid potential instability coming from optimizing an additional independent encoder.
>
> **3. Minimality and sufficiency**\
> The sufficiency is captured by the reconstruction terms, which are variational lower bounds on MIs between the representation and views. Combined with VIB, we aim for the learned representation that is minimal while sufficient, since the optimal solution of the reduction in the total correlation is achieved by learning a complete representation as we discussed in Section 2.1. We appreciate the comment and will revise the paper to make it more clear.
>
> **4. MMVAE doesn’t improve by adding more views**\
> This is because MMVAE adopts MoE as its joint representation encoder. While MoE ensures that individual experts are optimized, it is not fully capable of aggregating information across multiple views, which is a well studied limit of MoE as discussed by [34, 35]. We also discussed its limit in Section 2.4.
>
> **5. Decreased performance gap in Figure 3**\
> We suspect that the main reason for the decreased performance gap in the multi-view classification in Figure 3 is because 6 benchmark datasets used in this task are composed of hand-crafted features instead of raw observations, which allows us to extensively test all the models under our limited computational resource since the tasks are made easier. We selected these datasets since it was used in the landmark CPM-Net paper [50].
>
> Nonetheless, we remark that our method was able to obtain the representations that are consistently robust across all the tasks and datasets, including the ones composed of raw observations, such as PolyMNIST (Section 4.1.1) and MNIST-SVHN (in our response to reviewer 5CYt). All the experiments strongly imply that our method was the most reliable method to obtain the latent representation agnostic to the downstream tasks.
>
> **6. Other performance metrics besides accuracies**\
> Please note that classification and coherence accuracies are not the only metrics we used. For quantitative evaluation besides those accuracies, we reported FID score in Figure 1 to evaluate the quality of generated samples. In addition, we also compared our method to others with respect to reconstruction error in Figure 2 as another quantitative evaluation on the sample quality. For qualitative evaluation, in Section D.1, we extensively presented the result of conditional generation of baseline methods and ours. Furthermore, in Section D.2, we showcased how our method effectively aggregates information from given incomplete views. We will move some of these qualitative results to the main text in our revised paper.
>
> **7. Presentation issues including the Figure 4 and the order of modules**\
> We agree that Figure 4 should be moved to the main text since it describes the structure of our model, which will be reflected in the camera ready version. We thought that it is slightly more natural to explain first that how our method addresses unbalanced representation (Section 2.3) and then show how it can be extended to missing view scenarios (Section 2.4). We will change the order of 2 limitations of MVRL/PMVRL below line 93 so that readers are more likely to expect the current order of Section 2.3 and Section 2.4 in the camera ready version.

---

### Official Review · Reviewer_LwhH · 2021-07-14

**Rating:** 6
**Confidence:** 2

**Summary:**

This paper proposes to measure the quality of a multi-view generative model by the amount of total correlation among observations that is explained by the latent complete representation. While directly maximizing the reduction in total correlation is intractable, the authors derived a variational lower bound as a surrogate objective which encourages the model to learn a minimal sufficient representation and also regularizes the latent representation to be inferable from any view. The proposed formulation works well when some views are missing during training or testing. Hence it's suitable for partial multi-view representation learning. It also naturally calibrates the view-specific encoders to prevent the model heavily relying on only the most informative views. Empirical evaluation shows that the proposed method performs noticeably better than its baselines on most multi-view classification problems and multi-view translation tasks while achieves competitive results on a few others.

**Limitations And Societal Impact:**

The authors have mentioned potential societal impact of this work in the supplementary material.

**Main Review:**

Originality: the idea of using the reduction of total correlation as the objective of learning auto-encoder-like models has been studied in the literature. The authors have also cited related work in the submission. The originality of this paper comes from applying total correlation to multi-view representation learning and deriving a novel variational lower bound that is suitable for learning with partial observations.

Quality: the derivation of the proposed objective looks solid. But i haven't carefully checked the details in the supplementary material. Empirical study on the proposed method is pretty thorough. The paper can be strengthened if the authors can discuss about the computational complexity of their new method and the baselines. The choice of dimension of latent space is not covered in the main text either. It is not clear to me how much tuning effort the proposed method requires.

Clarity: this paper is well written in general but can be certainly polished further. On L136-137, the definition of encoder and decoder is confusing to me. The encoder, $r_{\psi}^v (z | o_v) = N(\mu_v, \sigma_v^2 I)$, is a Gaussian distribution that does not involve $o_v$, which feels unintuitive. The same for the decoder $q_{\psi}^v(o_v | z)$.

Significance: the novel variational lower bound and its application to partial multi-view representation learning appear to be a solid contribution to this domain. I am not very familiar with the state-of-the-art results on the related tasks. But, based on the experimental results, the proposed method performs well in practice and advances the best results further, though the amount of improvement varies by task.

**Time Spent Reviewing:**

8

---

> ### Author Response · Authors · 2021-08-10
> **Response to Reviewer LwhH**
>
> Thank you for your constructive feedback.
>
> **1. Computational complexity**\
> We appreciate the suggestion. We compared the computational complexity of ours and baseline methods as below:
>
> |          | &nbsp; MVAE [46]  &nbsp; | &nbsp; MMVAE [31] &nbsp; | &nbsp; mmJSD [34] &nbsp;| &nbsp; MoPoE [35] &nbsp; | &nbsp; Ours &nbsp; |
> | :------: | :-------: | :--------: | :--------: | :--------: | :----: |
> | Encoding |  $O(V)$   |   $O(V)$   |   $O(V)$   |  $O(2^V)$​  | $O(V)$ |
> | Decoding |  $O(V)$   | $O(KV^2)$​  |   $O(V)$   |   $O(V)$   | $O(V)$ |
>
> where $V$ is the nubmer of views and $K$ is the number of samples latent variables of each view used for optimizing IWAE bound.
>
> **2. Dimensions of latent variable**\
> Please note that all the comparing methods and ours used the same sized latent variables. Following the MoPoE-VAE paper [35], we used 512 dimensional latent variables when evaluating on PolyMNIST dataset. For 6 multi-view classification datasets, we followed settings in the CPM-Net paper [50], which are 128 dimensions for YaleB, CUB, and Handwritten datasets and 256, 150, 512 dimensions for ORL, PIE, Animal datasets respectively. For Caltech 101 dataset, we chose 100 dimensional latent variables since the effect of employing a bigger latent variable was marginal for all comparing methods.
>
> **3. Tuning effort**\
> Comparing to the baseline multi-view generative models [46, 31, 34, 35], we used the exact same hyperparameters, most of which are suggested by baseline method papers, except for $\alpha$​ introduced by our algorithm. Across all the experiments, setting $\alpha$ as one of 0.8, 0.9, 1.0 performs best in most cases. The effect of choosing different $\alpha$​’s is extensively experimented in Figure 2, Table 1, Table 2, and Table 3. Further details on hyperparameters can be found in Section F.
>
> **4. Clarification on view-specific encoder and decoder**\
> We appreciate your comment.  $\mu_v$​, $\sigma^{2}_{v}$​ are outputs of each view-specific encoder whose input is $o_v$​, so they are dependent on the input. Similarly, the $\hat{o}_v$​ is the output of each decoder, which is dependent on $z$​. We will make the notations clearer. We will move the Figure 4 in Section B, the structure of our model, to the main text as well for better presentation in the camera ready version.

---

> > ### Comment · Reviewer_LwhH · 2021-08-31
> > **Thank you for your responses**
> >
> > I would like to thank the authors for supplementing additional context in the responses. Indeed, Figure 4 in Section B is very helpful for understanding the architecture of the model. After reading the comments from other reviewer, I still think the proposed formulation is very interesting and like it a lot. But i also agree that the writing of this paper have lots of room for improvement. I will update my evaluation.

---

> > > ### Author Response · Authors · 2021-08-31
> > > **Additional response to reviewer LwhH**
> > >
> > > We appreciate your additional feedback.
> > > We are glad to hear that our formulation is interesting.
> > > We will revise our paper to improve the clarity, carefully considering all the suggestions from reviewers.
> > > Thank you.

---

### Official Review · Reviewer_5CYt · 2021-07-16

**Rating:** 5
**Confidence:** 4

**Summary:**

The paper proposes the variational-based generative approach to learn the multi-view (multi-modal) representation. The whole framework is motivated and derived from maximizing the total correlation  reduced by the representation, it focuses on learning a shared latent representation that is informative as well as succinct to capture the correlations among the given views (domains). Specifically, the framework is based on VAE framework with product-of-expert posterior encoder, and is trained via extended ELBO with additional regularization KL between joint encoder and view-specific encoder. The authors verify their proposed model on multi-view classification and translation. They also conduct experiments on partial multi-view representation learning where several views are missing in the training, and need to be inferred during testing.

**Ethical Concerns:**

No such concern.

**Limitations And Societal Impact:**

The limitation of the work could be described more clearly.

**Main Review:**

For originality:

 (+) the idea presented makes sense and interesting, though I feel the whole model is actually built on the MVAE with extra conditional VIB term as in eqn. 7. The paper also describes the difference with the existing relevant works.

For quality:

 (+) the authors consider various tasks (e.g., complete-views/partial views, translation/classification) for verification of their model.

 (-) however, it's better to also include some "standard" dataset for direct comparisons. For example, the existing baseline MMVAE use pair data from SVHN-MNIST for multi-domain translation/prediction (they also include MVAE performance), their models are well-tuned on such data for better performance and could be more illustrative.  Since the main difference is the training objective, if authors could use similar net structures to obtain results on such benchmark, that could easily give us direct and convincing comparisons.

 (-) perhaps the word "multi-view" is slightly misleading.  There are tons of multi-view representation learning papers in the literature that are mainly focused on face images under multiple viewpoints . This paper seems instead focused on multi-modal(domain) learning. Does the proposed model capable of handling face images under different view-points? Does the paper flexible enough to deal with raw pixels instead of extracted features on Multi-PIE data (for example)? It would be interesting to show the multi-view translation results on face (such as given one view point, use the learned network/representation to generate the face under novel view point).

For clarity:

 (-) the paper can be not so easy to follow. Such difficulty is not coming from the math derivation, but from the presentation of the paper. Quite a few symbols/notations are used without properly definition and descriptions beforehand. For instance: line 83, $r(z) \approx p_\theta(z)$, what is this $p_\theta(z)$? is it empirically marginal distribution based on observation or is it simply prior distribution of your decoder? If the latter, use $\theta$ indicates you have learnable prior? Another example would be $r_\phi$ in eqn. 7, when author write down eqn. 7, it didn't clearly show us what is the relationship between $p_\theta(z|o)$ and $r_\phi$. So have to go back and forth to check some notations in the previous sections. Changing the presentation would greatly improve the paper and make it more accessible.

For significance:

 (+) the new conditional VIB term might have some interesting properties and may be applicable for follow-up development.




**Time Spent Reviewing:**

3

---

> ### Author Response · Authors · 2021-08-10
> **Response to Reviewer 5CYt**
>
> We appreciate your constructive comments.
>
> **1. Additional experiment in MNIST-SVHN**\
> The table below summarizes the result of the experiment in MNIST-SVHN (MS) dataset averaged over 5 runs with seeds 0~4. The evaluation protocol, codebase, and hyperparameter settings follow the experiments in the MoPoE-VAE [35] on MNIST-SVHN-Text (MST) dataset, which augments to the original MS the text of digit IDs as the third modality. Please note that all the results below are from the baseline implementations in the MoPoE-VAE codebase whose VAE architectures dedicated to MNIST and SVHN are the same as the ones used by MMVAE. Lastly, we simply discarded the third modality.
>
> | % | Representation | Classification | &nbsp;&nbsp; &nbsp;&nbsp;(RC) &nbsp; &nbsp;&nbsp;| &nbsp;Coherent | Generation | &nbsp;(CG) |
> |:---:|:---:|:---:|:---:|:---:|:---:|:---:|
> |  | M (MNIST) | S (SVHN) | MS | Joint | M->S | S->M |
> | MVAE [46] | 87.38 | 58.23 | 87.37 | 42.98 | 56.65 | 35.63 |
> | MMVAE [31] | 72.83 | 60.89 | 66.89 | 42.45 | 26.76 | **74.94** |
> | mmJSD [34] | 88.58 | **81.44** | 93.81 | 12.73 | 22.33 | 65.44 |
> | MoPoE [35] | 82.48 | 70.62 | 87.96 | 44.95 | 21.23 | 72.38 |
> | Ours | **93.48** | 77.99 | **94.97** | **46.71** | **81.09** | 59.91 |
>
> where representation classification (RC) is measured by the accuracy of the single linear classifier trained on the latent representation as input (z from M only, z from S only, and z from jointly M and S), and coherent generation (CG) is measured by the accuracy of the pretrained CNN classifier whose input is the image generated by each model (e.g. Joint is measured from MNIST and SVHN images generated from the same z sampled from the prior distribution, and M->S is measured by SVHN images generated from MNIST images).
>
> Among 6 different evaluation results, our method outperforms baseline methods in 4 tasks (RC / MNIST and MS, CG / Joint and M->S) and performs competitive to the baselines in 1 task, RC / SVHN. Our method performs relatively poor only in CG / S->M (ranked 4th), the advantage of our method in M->S is much more noticeable. Comparing to strong baseline methods such as MMVAE, mmJSD, and MoPoE-VAE, our method shows more balanced performance in two different directions of CG, as measured in Joint, achieving the best average performance.
>
> **2. Novelty over MMVAE and MVAE**\
> Please note that differences between mmvae and our method are not only in the objective function but also in the choice of joint representation encoder. MMVAE chose MoE as their joint representation encoder to ensure that every view-specific encoder is optimized to infer the other views, potentially sacrificing the ability to aggregate information across all the given views. On the other hand, conditional VIBs in our formulation allow us not only to employ PoE to fully utilize the given views but also to enforce view-specific encoders to be optimized to infer missing modalities, which is one of the main advantages of our method over MVAE.
>
> **3. Terminologies of views, modalities, and domains**\
> We appreciate the comment. In our paper, we followed the conventional terminology in closely related works in the literature [14, 24, 41, 44, 47, 48, 50, 52], but any other choices among multi-views, multimodalities, and multiple domains can be suitable for our paper and our baseline methods, as long as there is a common latent representation that explains different views/modes/domains. We will add some clarification in introduction and Section 2 to make it more clear. Regarding multi-view translation of face images, we believe that our method can be naturally applied to this task by considering a face image taken from a certain camera angle as a view.
>
> **4. Notation and presentation**\
> $p_{\theta}(z)$ is a marginal distribution as defined in line 68. Although it is stated that $r_{\psi}(z|o_v)$ approximates $p_{\theta}(z|o_v)$ in section A.2 and $r(z)$ approximates $p_{\theta}(z)$ (which is actually held fixed in our implementation) in Section A.1, we agree that it would be better to state explicitly in the main text - we will revise them in the camera ready version.

---

### Official Review · Reviewer_XHEY · 2021-07-24

**Rating:** 6
**Confidence:** 2

**Summary:**

The paper motivates a variational objective (eq. (7)) for multi-view representation learning. The proposed objective is an alternative to the standard multi-view VAE objective, and should be better suited for extracting the correct representation when views are missing at random and when a subset of the views is overwhelmingly informative. They then propose optimising a convex combination of their proposed objective and the standard multi-view VAE objective.

**Limitations And Societal Impact:**

The authors address this in the supplementary material.

**Main Review:**

The problem tackled by the author seems well motivated, since multi-view representation learning in a partial setting (with missing views) is an important problem.
The paper is generally well structured and easy to read.

The definition of “complete representation” plays a crucial role in the paper, but is never stated formally. Adding a dedicated “Definition” paragraph would be useful for the reader. For example, as done in sec. 2.1 of:

Zhang, Changqing, et al. "CPM-Nets: Cross partial multi-view networks." (2019).

Lines 128-129 are very puzzling. After motivating and deriving objective (7), the authors say they “sometimes observed that the model is prone to overfitting”, and suggest instead optimising a convex combination of their proposed objective and the usual multi-view variational ELBO, favouring a minimally sufficient encoding.
The authors allude to a tradeoff between minimal sufficiency and completeness, but this is not extensively commented upon in the paper, despite apparently constituting a fundamental point for how it informs their final objective. The feeling is that something is missing here, both in theory (characterising this tradeoff, relating it to properties of the ground truth data generating process, etc.) and in experiments (e.g. the authors say “we observed that…”, which seems to refer to some unreported experiments).

I need to say, I am not an expert on the related literature (e.g. references [34] and [35] cited in the paper), so my confidence level in assessing its quality and novelty within that context is not extremely high. Overall I am under the impression that the paper, while being well motivated and written, is not very convincing in some parts, and that some key aspects would require more elaboration and possibly a more rigorous treatment, as detailed above.

Typos and minor comments:

References [12] and [13] are the same?

Line 191: “variable.Deep” — space missing after dot

The captions for figures 1, 2 and 3 could be made more informative.

**Update** I will increase my evaluation to 6 based on the authors' answer.

**Time Spent Reviewing:**

4

---

> ### Author Response · Authors · 2021-08-10
> **Response to Reviewer XHEY**
>
> Thank you for your thoughtful feedback.
>
> As reviewer pointed out, we follow the definition of a complete representation in CPM-Nets [50]. We will include a formal definition in the main text of the paper to improve readability. We will also incorporate the editorial suggestions made by the reviewer in the final version of the paper.
>
> **Clarification on our main objective function**\
> Firstly, please note that both Eq.(5) (derived using VIB) and Eq.(7) (derived using conditional VIB) are two different derivations of lower bounds on Eq.(4), aiming to learn a complete representation with different regularizations. Thus, $\alpha$ in Eq.(8), the convex combination weight of Eq.(5) and Eq.(7), controls the tradeoff not between completeness and minimal sufficiency but between VIB and conditional VIBs. For successful multi-view representation learning, it is desirable to enjoy complementary benefits of them: conditional VIBs are essential for learning balanced representations and handling missing views while VIB is useful for avoiding overfitting. We can observe its effectiveness by comparing Figure 2.a and Figure 2.b, which summarize the results of training with complete and incomplete observations respectively. When the complete dataset is given, $\alpha=1.0$ (conditional VIB only) outperforms $\alpha=0.8$ (0.8 * conditional VIB + 0.2 * VIB). In contrast, given the dataset with missing views by making the training data more scarce, $\alpha=0.8$ outperforms $\alpha=1.0$. The effect of $\alpha$ is experimented more in Section C.1.

---

> > ### Comment · Reviewer_XHEY · 2021-08-30
> > **Thanks for your answer**
> >
> > I thank the authors for their clarification and pointing out my previous misunderstanding. I still find the formulation at lines 128-129 unclear: at that point in the paper, where the authors have not yet presented any experimental results and are laying out the derivation of their proposed objective functions, the sentence _"we sometimes observed that the model is prone to overfitting"_ seems out of place or poorly worded. Generally, I agree with reviewer `5CYt` that the clarity of some parts of the paper could be improved. I will nevertheless update my previous evaluation based on the author's answer.

---

> > > ### Author Response · Authors · 2021-08-31
> > > **Additional response to reviewer XHEY**
> > >
> > > We appreciate your additional feedback.
> > > We will carefully reflect all the suggestions from reviewers to improve the clarity in our paper.
> > > Thank you.

---

### Author Response · Authors · 2021-08-19
**Dear reviewers and AC**

Dear reviewers and AC,

We sincerely appreciate reviewers’ effort to make fair assessments of our work and provide constructive feedback.
We hope that our response has resolved all the concerns raised by reviewers.
In addition, we would be more than happy to actively participate in the rolling discussion to address any remaining concerns.

Best regards, Authors of paper 4419.

---

### Decision · Program_Chairs · 2021-09-27

**Decision:**

Accept (Poster)

**Comment:**

This paper proposes a total correlation objective within VAE framework for multi-view representation learning, and the proposed method can handle situations with missing views.

The main concern of the reviewers is with the presentation and clarity.

The overall assessment is that this is an incrementally novel contribution with incrementally improved experimental results.